# How Many Different Outputs Can a Transformer Generate?

**Maxime Meyer** [* 1 2]   **Mario Michelessa** [* 2 3]   **Caroline Chaux** [2 4]   **Vincent Y. F. Tan** [1 5]

## Abstract

We study how we can leverage only a handful of characteristics of a transformer's architecture to closely predict the number of different sequences it can output, both qualitatively and quantitatively. We provide an upper bound depending on the length of the prompt, which we show empirically to be tight up to a factor less than 10, across architectures and model sizes. Our analysis also provides a theoretical explanation for previously observed empirical failures of transformers on simple sequence tasks—such as copying and cramming. Formally, we prove that (i) the maximal length of accessible sequences (those that the transformer can output for some prompt) grows linearly with the prompt length, (ii) beyond a critical threshold, the proportion of accessible sequences decays exponentially with sequence length, and (iii) the linear coefficient relating prompt length to accessible sequence length admits a theoretical upper bound. Notably, these results hold even with unbounded context and computation time.

## 1. Introduction

Transformers (Vaswani et al., 2017) have become the dominant architecture in modern deep learning, achieving state-of-the-art results across domains such as natural language processing (Wolf et al., 2020) and computer vision (Dosovitskiy et al., 2021; Zhao et al., 2021; Zhai et al., 2022). Despite their success, these models fail on surprisingly simple tasks—such as copying or repeating sequences (Jelassi et al., 2024)—once the input exceeds a certain length. These

failures persist even for large models specifically trained on those tasks, suggesting that they may reflect intrinsic architectural constraints.

This paper characterizes such constraints by identifying structural limitations that hold for all transformers, independently of model size, training data, or available computation:

- We show that every transformer can output only a finite set of output sequences. That is, most sequences are fundamentally inaccessible: no choice of prompt, context length, or inference strategy can cause the model to generate them. This impossibility result provides a principled explanation for well-documented failures of transformers on simple algorithmic tasks, such as exact copying.

- Moreover, we show that for a fixed prompt length, the proportion of accessible sequences of length $n$ decays exponentially fast with $n$ beyond a model-dependent threshold. This phenomenon explains a puzzling empirical behavior observed in practice, where transformers perform nearly perfectly up to a critical length and then fail abruptly rather than gradually.

- Finally, we derive an explicit formula to predict those thresholds as functions of the transformer architecture being used. Remarkably, this theoretical result closely matches empirical performance observed across a wide range of transformer architectures and model sizes.

To establish these results, we analyze how the finite precision and the geometry of the internal representations of a transformer constrain the set of sequences it can represent or generate. We formalize this constraint through the notion of *accessible sequences*, defined as outputs that can arise from some input prompt (Section 3). The main theoretical analysis, which yields the results stated above, is conducted in Section 4. We also devise experiments that validate our theoretical predictions in Section 5[1].

*Remark* 1.1 (Broader Applicability of Our Results). Our bounds extend beyond transformers and apply to any architecture with bounded internal representations and finite

---

[*]Equal contribution  [1]Department of Mathematics, National University of Singapore, Singapore, 117543 [2]IPAL, IRL2955, Singapore [3]School of Computing, National University of Singapore, Singapore, 117543 [4]Aix Marseille Univ, CNRS, I2M, Marseille, France [5]Department of Electrical and Computer Engineering, National University of Singapore. Correspondence to: Maxime Meyer <maxime.meyer@u.nus.edu>.

*Proceedings of the 43$^{rd}$ International Conference on Machine Learning*, Seoul, South Korea. PMLR 306, 2026. Copyright 2026 by the author(s).

---

[1]Code is available at github.com/mario-michelessa/transformers_accessibility

precision. This may include state-space models such as Mamba (Gu & Dao, 2024). However, the possibility of diverging hidden states (Lu et al., 2026) makes the analysis more challenging, and we leave this direction for future work.

## 1.1. Related Works

**Approximation Theory of Transformers.** The representational power of transformers has been extensively explored through the lens of approximation theory. In contrast, we study how finite numerical precision constrains this expressivity—showing that, as rounding errors accumulate, even simple functions such as the identity map become difficult to approximate in practice. Yun et al. (2020a) established the first universal approximation property for continuous functions, later extended to sparse attention transformers (Yun et al., 2020b) and sparse target functions (Edelman et al., 2022). Subsequent works refined these results for one-layer architectures (Kajitsuka & Sato, 2024; Jiang & Li, 2024) and examined approximation under compact output constraints (Kratsios et al., 2022). Architectural factors such as relative positional encodings (Luo et al., 2022) have also been analyzed. A complementary line of work investigates approximation properties under prompt tuning (Wang et al., 2023; Hu et al., 2025). Among those, Meyer et al. (2025) uses a proof strategy similar to ours, leveraging a packing number-type argument.

**Empirical Analyses of Sequence Accessibility.** Empirical studies have also examined the ability of transformers to reproduce or memorize sequences. Barbero et al. (2024) showed that transformers struggle to reliably copy long sequences, but their analysis focuses on highly synthetic settings—sequences differing by only a few tokens or alternating between two symbols (*e.g.* 0 and 1). In contrast, our work studies the capacity of transformers to generate, and therefore copy, arbitrary sequences. In this line of work, Jelassi et al. (2024) showed that even transformers trained on the specific task of copying struggle to reliably execute this task as the sequence length increases. The work most closely related to ours is Kuratov et al. (2025), who introduced the "cramming" procedure as an exact test of accessibility. For any target sequence $[\mathbf{x}_1, \ldots, \mathbf{x}_n]$, a set of trainable memory vectors $[\mathbf{mem}] = [\mathbf{y}_1, \ldots, \mathbf{y}_m]$ is optimized so that the model generates $[\mathbf{x}_1, \ldots, \mathbf{x}_n]$ when conditioned on $[\mathbf{mem}]$. This approach effectively searches over all possible prompts—including highly unnatural ones—to determine whether a sequence is accessible. Cramming thus provides an operational definition of accessibility: a sequence is accessible if and only if a corresponding encoding $[\mathbf{mem}]$ exists. Our work provides the first rigorous theoretical explanation for the empirical findings in Kuratov et al. (2025), and extends the experimental analysis beyond

the original setting. We describe this further in Section 5.

**Mean-Field Transformers.** To analyze prompts of arbitrary length, it is useful to interpret a transformer as a mapping between probability measures rather than finite-length sequences, following the mean-field formulation introduced by Sander et al. (2022) and defined in Section 2.2. This perspective has been further developed in several directions. Castin et al. (2024) employed it to study the Lipschitz continuity of transformers, while Furuya et al. (2025) extended the approximation theory of transformers to the space of probability distributions. More recently, Meyer et al. (2025) leveraged the mean-field framework to characterize the memory limitations of transformers. Our analysis builds upon this formulation to derive accessibility bounds that hold independent of the prompt.

**Representations of the Embedding Space.** Prior work has visualized and probed LLM embedding spaces in three main ways. First, transformer representations have been shown to be highly anisotropic, with a large fraction of the variance concentrated along a small number of directions (Mu & Viswanath, 2018; Ethayarajh, 2019). Second, probing methods have shown that certain syntactic and semantic properties are linearly recoverable from transformer hidden representations (Alain & Bengio, 2016; Hewitt & Manning, 2019; Kornblith et al., 2019). Third, decoding intermediate activations by projecting them through the decoder matrix to logits reveals layerwise sharpening of next-token beliefs (Nostalgebraist, 2020; Belrose et al., 2023). Inspired by this decoding view, we use the same decoder matrix to find hyperplanes that partition the embedding space into next-token argmax cells in Section 3.

**Expressive Power of Transformers.** Complementary to the line of work on approximation theory, the expressivity of transformers can also be studied in terms of the formal languages they can express (Strobl et al., 2024). Transformers have been shown to represent Turing machines (Wei et al., 2022) and arbitrary computer programs (Giannou et al., 2023). The impact of chain-of-thought reasoning (Merrill & Sabharwal, 2024) and precision (Chiang, 2025) have also been studied. Most relevant to our work is the work of Huang et al. (2025), who prove the impossibility of copying arbitrary sequences for transformers with absolute positional encodings and under some idealized assumptions. We extend those results by removing those assumptions and providing both a qualitative and quantitative analysis of the performance of transformers in the copying task.

## 1.2. Notations

Bold lowercase letters (e.g., $\mathbf{x}$) denote vectors, and bold uppercase letters (e.g., $\mathbf{W}$) denote matrices. For a matrix

$\mathbf{W}$, we write $\mathbf{W}_{i,j}$, $\mathbf{W}_{i,:}$, and $\mathbf{W}_{:,j}$ to refer to its $(i,j)$-th element, $i$-th row, and $j$-th column, respectively. We can use $i = -1$ and $j = -1$ to denote the last row and column, respectively. The concatenation of $k$ matrices with the same number of rows $d$, $\mathbf{W}^1 \in \mathbb{R}^{d \times m_i}, \ldots, \mathbf{W}^k \in \mathbb{R}^{d \times m_k}$ is denoted as $[\mathbf{W}^1, \ldots, \mathbf{W}^k] \in \mathbb{R}^{d \times \sum_{i=1}^{k} m_i}$.

We write $\sigma$ to denote the softmax function. The rectified linear unit is defined as $\mathrm{ReLU}(v) = \max(v, 0)$, where the maximum is applied elementwise. We use $|\mathcal{A}|$ to denote both the cardinality of a finite set, and the volume of a set $\mathcal{A} \subset \mathbb{R}^n$ for some $n \in \mathbb{N}$ (we take $\mathbb{N}$ to be the set of strictly positive integers). We denote by $\mathcal{V} = \{t_1, \ldots, t_{|\mathcal{V}|}\}$ the vocabulary (that is the set of all distinct tokens) and $d$ the embedding dimension. The probability simplex over the vocabulary is denoted as $\Delta^{\mathcal{V}}$.

Throughout, $\|\cdot\|$ denotes a norm, which by default may be taken as the standard $\ell_\infty$ or $\ell_p$ norm for vectors. Unless otherwise specified, our results hold for all of these norms. Let $\Pi(\mu, \nu)$ be the set of all joint distributions with marginals $\mu$ and $\nu$; this is also called the set of *couplings* of $\mu$ and $\nu$. When considering distributions, we use the Wasserstein distance

$$W_q(\mu, \nu) = \left( \inf_{\pi \in \Pi(\mu, \nu)} \int \|x - y\|^q \, \mathrm{d}\pi(x, y) \right)^{1/q}.$$

This distance is further discussed in Appendix D.

## 2. The Transformer Architecture

### 2.1. Standard Transformers

We consider transformer networks (Vaswani et al., 2017) composed of repeated self-attention and MLP layers, together with an embedding layer and a final linear projection.

**Definition 2.1** (One-Hot Representation). Let $\mathcal{V} = \{t_1, \ldots, t_{|\mathcal{V}|}\}$ denote the vocabulary. For a token $t_i \in \mathcal{V}$, the *one-hot representation* of $t_i$ is the canonical vector

$$\mathbf{c}_i = (\mathbf{1}_{i=j})_{j \in [|\mathcal{V}|]} \in \mathbb{R}^{|\mathcal{V}|}.$$

**Definition 2.2** (Embedding Layer). Let $\mathbf{E} \in \mathbb{R}^{d \times |\mathcal{V}|}$ be the embedding matrix. For a sequence of $m$ tokens $\mathbf{t} = (t_{i_1}, \ldots, t_{i_m}) \in \mathcal{V}^m$, the embedding layer maps the one-hot representation $[\mathbf{c}_{i_1}, \ldots, \mathbf{c}_{i_m}] \in \mathbb{R}^{|\mathcal{V}| \times m}$ to

$$\mathbf{X_t} = \mathbf{E}[\mathbf{c}_{i_1}, \ldots, \mathbf{c}_{i_m}] \in \mathbb{R}^{d \times m}.$$

**Definition 2.3** (Transformer with Hard Prompt Inputs). An $l$-layer transformer with vocabulary $\mathcal{V}$ and unembedding matrix $\mathbf{F} \in \mathbb{R}^{|\mathcal{V}| \times d}$ is the mapping

$$\tau : \bigcup_{m=1}^{+\infty} \mathcal{V}^m \longrightarrow \Delta^{\mathcal{V}},$$

defined as

$$\tau(\mathbf{t}) = \sigma\big( \mathbf{F}(\mathbf{X}_l)_{:,-1} \big), \quad \text{where}$$

$\mathbf{X}_l = \mathrm{L}(\mathbf{X_t})$ is the output of the final transformer layer,

$\mathbf{X_t}$ is the embedded input with positional encodings.

The only characteristic of the transformer layers L required in this paper is that they keep the embedding space bounded. The formal proof and empirical validation, together with the formal definition of L are deferred to Appendix A.

A transformer can also be seen as mapping soft prompt inputs from $\bigcup_{m=1}^{+\infty} \mathbb{R}^{d \times m}$ (which may or may not correspond to sequences of tokens).

**Definition 2.4** (Transformer with Soft Prompt Inputs). An $l$-layer transformer with embedding dimension $d$ is the mapping

$$\tau : \bigcup_{m=1}^{+\infty} \mathbb{R}^{d \times m} \longrightarrow \Delta^{\mathcal{V}},$$

defined as

$$\tau(\mathbf{X}) = \sigma\big( \mathbf{F}\big( \mathrm{L}(\mathbf{X}) \big)_{:,-1} \big).$$

We consider Definition 2.4 in our analysis. However, since a transformer with soft prompt inputs is more expressive than its hard prompt counterpart, all of our results directly extend to Definition 2.3.

### 2.2. Mean-Field Transformers

To handle arbitrary prompt lengths, we can interpret transformers as maps over probability measures (Sander et al., 2022; Geshkovski et al., 2023). This is possible because of two key properties of attention (and hence of a transformer as defined in Definition 2.4):

- Attention layers are permutation equivariant.

- For any attention block L and input $\mathbf{X}$, $\mathrm{L}(\mathbf{X})_{:,i} = \mathrm{L}([\mathbf{X}, \mathbf{X}])_{:,i}$.

We can thus replace every sequence $\mathbf{X}$ of length $m$ with its associated empirical measure

$$\mathrm{M}(\mathbf{X}) := \frac{1}{m} \sum_{i=1}^{m} \delta_{\mathbf{X}_{:,i}},$$

so that a transformer can be viewed as a transformation of probability measures supported on $\mathbb{R}^d$, independent of the ordering of tokens. This is called the mean-field extension of transformers, and we describe it further in Appendix B.

*Remark* 2.5 (Positional Encodings and Masked Attention). The mean-field framework can also be extended to most types of positional encodings and to masked attention, as discussed in Appendix B.

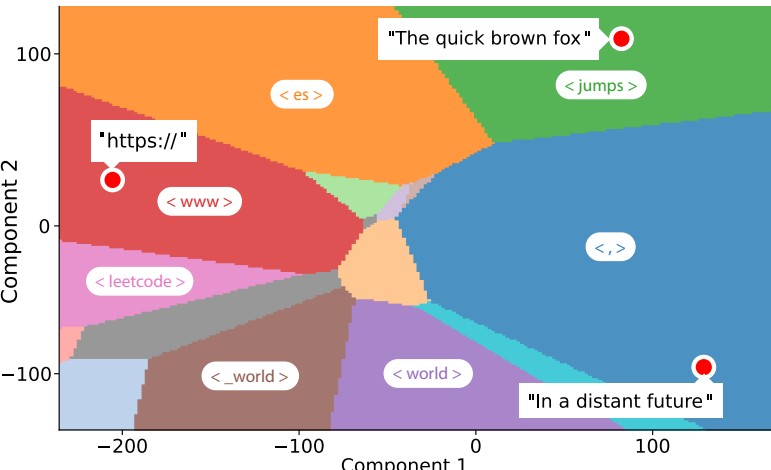

*Figure 1.* Plane cut of the embedding space $\mathcal{E}$ of Qwen-2 (0.5B) (Yang et al., 2024), passing through the final token embeddings of "The quick brown fox", "https://", and "In a distant future" (red markers). Colors encode, for each pixel, the most probable next token via the model's output projection (decoder readout). This induces regions $\mathcal{E}_t$ annotated by $< t >$. Only tokens whose regions have the largest areas are annotated for readability. The three embeddings fall respectively in $\mathcal{E}_{\text{jumps}}$, $\mathcal{E}_{\text{www}}$, and $\mathcal{E}_,$, *i.e.* the predicted next tokens are $<\text{jumps}>$, $<\text{www}>$, and $<,>$.

## 3. Representing the Embedding Space

### 3.1. Partitioning the Embedding Space by the most Likely Next Token

Let us fix a transformer $\tau$ of embedding dimension $d$. We follow empirical evidence (Brody et al., 2023) in assuming that the embedding support—that is the fraction of the embedding space being utilized by $\tau$—of any token at any layer of $\tau$ is a subset of $B^d(0, r) := \{\mathbf{x} \in \mathbb{R}^d, \|\mathbf{x}\| < r\}$. We refer to $r$ as the embedding radius. Then we can partition the embedding support of the final tokens at the last layer $\mathcal{E}$ depending on the most likely next token $\mathcal{E} = \bigcup_{t_i \in \mathcal{V}} \mathcal{E}_i$, where for each token $t_i \in \mathcal{V}$,

$$\mathcal{E}_i = \{\mathbf{x} \in \mathcal{E}, \forall j \in [|\mathcal{V}|], (\mathbf{Fx})_i \geq (\mathbf{Fx})_j\},$$

corresponds to the set of last embedding vectors that most likely lead to the next token $t_i$.

Figure 1 shows an illustration of the regions corresponding to the most likely next token in the embedding space. Similarly, we can also study the most likely next two tokens, next three tokens, *etc*. As we consider the most likely next $n$ tokens, the regions become exponentially small with $n$. At some point, these regions therefore become inaccessible for a transformer. This is the intuition behind our analysis, which we formalize in the remainder of the paper.

*Remark* 3.1. An important thing to note is that in order to predict the next $n$ tokens, we need to consider the embedding of the whole prompt. This contrasts with simply predicting the next token, where only the embedding of the last token at the last layer is needed. This is why in the following, we denote $\mathcal{E} \subset \mathbb{R}^{d \times m}$ the embedding space of the whole prompt of size $m$.

### 3.2. Accessing these regions through Prompts

A natural factor affecting which sequences of tokens can be accessed by a transformer is the prompt length: if the prompt length is limited, a transformer is able to access fewer outputs than if the prompt length is arbitrarily large. To study this, we introduce a second partition of the embedding space.

For every sequence of tokens $\mathbf{t} = (t_{i_1}, \ldots, t_{i_n}) \in \mathcal{V}^n$ and prompt length $m \in \mathbb{N}$, we can write $\mathcal{E}_{\mathbf{t}}^m$ the set of soft prompts of length $m$ that lead to the sequence $\mathbf{t}$ under greedy decoding, that is

$$\mathcal{E}_{\mathbf{t}}^m = \left\{ \mathbf{X} \in B^{d \times m}(0, r) \colon \forall j \in [n], \right.$$

$$\left. \operatorname{argmax}\left( \mathbf{FL}\Big([\mathbf{X}, \mathbf{Et}_{i_1}, \ldots, \mathbf{Et}_{i_{j-1}}]\Big)_{-1} \right) = i_j \right\},$$

Then $B^{d \times m}(0, r) = \bigcup_{\mathbf{t} \in \mathcal{V}^n} \mathcal{E}_{\mathbf{t}}^m$, where we can replace $\mathcal{E}_{\mathbf{t}}^m$ with its closure $\overline{\mathcal{E}_{\mathbf{t}}^m}$ depending on how we handle cases where the final vector admits multiple argmax values. In the following, we simply write $\mathcal{E}_{\mathbf{t}}$, as $m$ is already implied through $\mathbf{t}$.

*Remark* 3.2 (Other Decoding Strategies). A direct corollary of our results is that if a sequence is not accessible under greedy decoding, then it cannot be generated with probability greater than 50% under any stochastic decoding strategy (e.g., beam search or Top-$K$ sampling). As a result, our findings for the copying task extend to discrete decoding strategies, as they provide an upper bound on the success probability.

# 4. The Sequences that are Accessible to a Transformer

With the partitions of the embedding space defined in Section 3, we are now equipped to formalize our problem. Specifically, we aim to study the set of sequences that are accessible to a given transformer $\tau$. A given sequence $\mathbf{t}$ is said to be accessible if there exists a prompt for which the output under greedy decoding is $\mathbf{t}$.

**Definition 4.1** (Accessible Sequence). A sequence of tokens $\mathbf{t} \in \mathcal{V}^n$ is said to be accessible by a transformer $\tau$ of precision $\varepsilon$ with prompt length $m$ if there exists a sequence $\mathbf{X} \in \mathbb{R}^{d \times m}$ such that $B(\mathbf{X}, \frac{\varepsilon}{2}) \subset \mathcal{E}_{\mathbf{t}}^m$. A sequence is said to be accessible if it is accessible with prompt length $m$ for some $m \in \mathbb{N}$.

This notion of accessible sequences enables us to formally state our contributions.

- In Section 4.1, we show that the maximal length $n$ of accessible sequences scales linearly with the prompt length $m$.

- In Section 4.2, we show that for every transformer $\tau$, there exists a certain threshold $N \in \mathbb{N}$, above which almost all sequences of length $n \geq N$ are not accessible by $\tau$.

Our proofs require the notion of packing number (Vershynin, 2026), which quantifies how many disjoint balls of a given radius one can pack in a space.

**Definition 4.2** (Packing Number). Let $(T, \|\cdot\|)$ be a vector space and let $\varepsilon > 0$.

The $\varepsilon$-*packing number* of $T$, denoted $\mathcal{P}(T, \|\cdot\|, \varepsilon)$, is the maximal number of disjoint open balls of radius $\frac{\varepsilon}{2}$ that can be placed in $T$, or equivalently, the largest cardinality of an $\varepsilon$-separated subset of $T$. That is,

$$\mathcal{P}(T, \|\cdot\|, \varepsilon) = \max\{M \in \mathbb{N} \colon \exists (x_i)_{i \in [M]} \in T^M,$$
$$\forall i \neq j, \|x_i - x_j\| > \varepsilon\}.$$

We conduct further analysis on the packing number of the spaces we are studying in Appendix C.

## 4.1. Finite Prompt Length

Our study of the finite prompt length setting is based on the trivial assumption that any transformer has limited precision. That is, it cannot distinguish between two inputs that are too close to each other.

**Assumption 4.3.** [Finite Precision] Every transformer $\tau$ admits a finite precision $\varepsilon > 0$. That is $\mathbb{R}^d$ can be partitioned into axis-aligned cubes of side length $\varepsilon$, and $\tau$ is constant within each cube:

$$\tau(\mathbf{X}) = \tau\left(\left\lfloor \frac{\mathbf{X}}{\varepsilon} \right\rfloor \varepsilon\right) \quad \text{for all } \mathbf{X}.$$

*Remark* 4.4 (Extension to Finite $\ell_p$ Precision). We used the $\ell_\infty$ norm in Assumption 4.3 as it is the most intuitive. Note that this directly generalizes to $\ell_p$ precision as $\|\mathbf{X}\|_\infty \leq \|\mathbf{X}\|_p$ implies that a transformer with precision $\varepsilon$ is also constant on each ball of a grid of $\ell_p$ balls of radius $\varepsilon$.

Under Assumption 4.3, we show that the maximal length $n$ of accessible sequences scales linearly with the prompt size $m$.

**Theorem 4.5.** *A transformer of precision $\varepsilon$ and embedding radius $r$ with prompt length $m$ can only access a finite number of distinct sequences: $\left(1 + \frac{2r}{\varepsilon}\right)^{dm}$.*

*Proof.* Let us fix a transformer $\tau$ with embedding radius $r$ and precision $\epsilon$. Clearly, this transformer can only distinguish between at most $\mathcal{P}(B^{d \times m}(0, r), \|\cdot\|, \varepsilon)$ different inputs of size $m$. Hence, the number of distinct accessible sequences of tokens with prompt length $m$ is upper bounded by $\mathcal{P}(B^{d \times m}(0, r), \|\cdot\|, \varepsilon)$, and thus by $\left(1 + \frac{2r}{\varepsilon}\right)^{dm}$ [Proposition C.2]. $\square$

**Corollary 4.6.** *When* $n > \left(d \dfrac{\ln(1 + \frac{2r}{\varepsilon})}{\ln(|\mathcal{V}|)}\right) m$, *there exist some sequences of length at most $n$ that are not accessible by a transformer of precision $\varepsilon$ and embedding radius $r$ with prompt length $m$. Moreover, the proportion of accessible token sequences decays exponentially fast with $n$ when it exceeds the previous threshold, with rate* $\dfrac{(1 + \frac{2r}{\varepsilon})^{dm}}{|\mathcal{V}|^n} \in O(\frac{1}{|\mathcal{V}|^n})$.

*Proof.* Combine Theorem 4.5 with the fact that the number of distinct possible output token sequences of size $n$ is given by $|\mathcal{V}|^n$. $\square$

*Remark* 4.7. Although there is no strict maximal length beyond which no sequence is accessible, the proportion of accessible sequences decreases exponentially once the critical length is exceeded. In practice, this rapid decay renders the distinction negligible: beyond that threshold, almost all sequences become effectively inaccessible.

Note that we made two crucial assumptions in this section:

- the embedding support is a fully used ball of radius $r$.

- each token sequence $\mathbf{t}$ corresponds to a region $\mathcal{E}_{\mathbf{t}}$ of equal volume in the embedding space.

Those assumptions consider the worst case setting, in the sense that a model not satisfying those assumptions has a smaller (and hence tighter) theoretical upper bound on the slope. Indeed, if the embedding support does not fully occupy the ball of radius $r$, then it is smaller and $\tau$ can access less sequences. And if some tokens correspond to smaller regions $\mathcal{E}_t$, then those tokens are harder to access, and the threshold for which some sequences are inaccessible is smaller.

We describe in Section 5 how we can relax these assumptions by better approximating the embedding support $\mathcal{E}$ (Section 5.2) and by empirically estimating the distribution of cell volumes $|\mathcal{E}_t|$ (Section 5.3). A more formal analysis is provided in Appendix F.

### 4.2. Arbitrary Prompt Length via the Mean-Field Generalization

Our goal in this section is to obtain an upper bound on the length of the sequences accessible by a transformer, independent of the input length. Notice that there is a straightforward way to do this if Assumption 4.3 can be generalized to Wasserstein distance.

**Assumption 4.8.** Every transformer $\tau$ admits a finite Wasserstein precision. That is, the space of inputs can be partitioned into finitely many $W_q$-balls of radius $\varepsilon$, and $\tau$ is constant within each ball.

Formally, there exist $\varepsilon > 0$ and $q \geq 1$ such that

$$\tau(\mathbf{X}) = \tau\left(\underset{\mathbf{Z} \in \mathcal{G}_\varepsilon}{\arg\min}\, W_q\big(\mathrm{M}(\mathbf{X}), \mathrm{M}(\mathbf{Z})\big)\right) \quad \text{for all } \mathbf{X},$$

where $\mathcal{G}_\varepsilon$ is a finite $\varepsilon$-net of reference sequences under $W_q$.

Under this assumption (the validity of which is further discussed in Appendix D), we show that for every transformer $\tau$, there exists a certain threshold $N \in \mathbb{N}$, above which almost all sequences of length $n \geq N$ are not accessible by $\tau$.

**Theorem 4.9.** *A transformer of precision $\varepsilon$ and embedding radius $r$ can only access a finite number of distinct sequences:* $\left(e + \frac{e\,(2r)^q}{\varepsilon^q}\right)^{\left(1 + \frac{2r}{\varepsilon}\right)^d}$.

*Proof.* Under Assumption 4.8, let us fix a transformer $\tau$ with embedding radius $r$ and precision $\epsilon$. Clearly, this transformer can only distinguish between at most $\mathcal{P}(\mathcal{G}, W_q, \varepsilon)$ different inputs, with $\mathcal{G} := \mathrm{Im}(\mathrm{M}) = \{\mathrm{M}(\mathbf{X}) \colon \mathbf{X} \in \mathbb{R}^{d \times m}, m \in \mathbb{N}\}$ the set of discrete probability measures on the token embeddings of dimension $d$ as defined in Section 2.2. Hence the number of distinct accessible sequences of tokens is upper bounded by $\mathcal{P}(\mathcal{G}, W_q, \varepsilon)$, and thus by $\left(e + \frac{e\,(2r)^q}{\varepsilon^q}\right)^{\left(1 + \frac{2r}{\varepsilon}\right)^d}$ [Proposition C.3]. $\square$

**Corollary 4.10.** *When $n > \left(1 + \frac{4r}{\varepsilon}\right)^{d} \frac{\ln\left(e + \frac{e\,(2r)^q}{\varepsilon^q}\right)}{\ln(|\mathcal{V}|)}$, there exist some sequences of length at most $n$ that are not accessible by a transformer of precision $\varepsilon$ and embedding radius $r$. Moreover, the proportion of accessible token sequences decays exponentially fast with $n$ when it exceeds the previous threshold, at rate* $\frac{\left(e + \frac{e\,(2r)^q}{\varepsilon^q}\right)^{\left(1 + \frac{4r}{\varepsilon}\right)^d}}{|\mathcal{V}|^n} \in O\left(\frac{1}{|\mathcal{V}|^n}\right)$.

*Proof.* Combine Theorem 4.9 with the fact that the number of distinct possible output token sequences of size $n$ is given by $|\mathcal{V}|^n$. $\square$

*Remark* 4.11 (Arbitrary Prompt Length via Elementary Operations). Contrary to Assumption 4.3, Assumption 4.8 is non-trivial. We show in Appendix E that it can be replaced by a much weaker assumption, pertaining to what we call elementary operations.

## 5. Experiments

Corollary 4.6 makes three concrete predictions. First, for a fixed prompt budget $m$, accessibility stays high up to a length threshold $n^\star(m)$, and then the fraction of accessible sequences decays exponentially fast as $n$ increases. Second, this threshold grows linearly with the prompt budget, i.e., $n^\star(m) = Cm$ for some $C > 0$. Third, the corollary provides an explicit upper bound on $C$.

We first test the first two predictions through the cramming task (Kuratov et al., 2025) in Section 5.1: we measure whether a frozen transformer can generate a target sequence using optimized soft prompts. We next refine the theoretical upper bound and compare it to the empirical slope (Section 5.2, 5.3).

Finally, we illustrate the saturation regime predicted in Corollary 4.10 using the (easier to compute) copying task.

### 5.1. The Cramming Task

We want to test whether a target sequence is accessible, i.e., whether there exists some prompt that makes a model generate it. Directly searching over discrete prompts is intractable, so we optimize a length-$m$ soft prompt and check for an exact match. Given a pretrained transformer $\tau$ and a target token sequence $x_{1:n}$, we say the sequence is accessible with prompt length $m$ if there exists a sequence of $m$ soft prompt vectors $Y \in \mathbb{R}^{d \times m}$ such that greedy decoding from $Y$ generates exactly $x_{1:n}$. To search for such a prompt, we use the cramming setup of Kuratov et al. (2025): we freeze the model weights and optimize only $Y$ to maximize the likelihood of the target under teacher forcing.

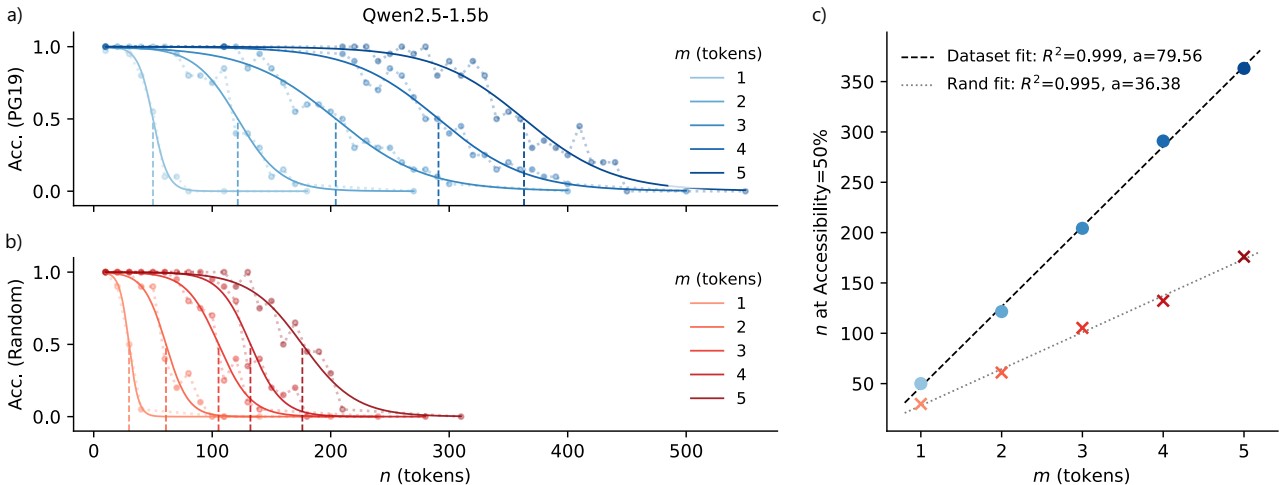

*Figure 2.* Mean accessibility for a) PG19 and b) random target sequences of length $n$ as a function of the number of trainable memory vectors $m$. For each $m$, we fit a sigmoid (solid) and mark $n_{50}$ where the fit crosses 0.5 (vertical dashed line). (c) $n_{50}(m)$ for PG19 (blue) and random (red), with linear fits (dashed).

Concretely, we minimize the cross-entropy

$$\mathcal{L}(Y; x_{1:n}) = -\sum_{i=1}^{n} \log p_\tau(x_i \mid [Y, x_{1:i-1}]), \quad (1)$$

After optimization, we evaluate accessibility by generating exactly $n$ tokens greedily from the learned prompt $Y$:

$$\hat{x}_i = \arg\max_{t \in \mathcal{V}} p_\tau(t \mid [Y, \hat{x}_{1:i-1}]), \qquad i = 1, \ldots, n, \quad (2)$$

and we count a success if $\hat{x}_{1:n} = x_{1:n}$. For each $(n, m)$ we estimate accessibility as the mean success rate over 20 target strings. We consider two target sources: i) contiguous token spans of length $n$ sampled from PG19 (Rae et al., 2020) using each model's tokenizer, and ii) random strings formed by sampling tokens uniformly i.i.d. from a subset of the model vocabulary.

We test whether, for fixed $m$, success stays high up to a length threshold and then drops quickly as $n$ increases; and if this threshold scales roughly linearly with $m$. We run the same procedure across several model families and sizes: Pythia (160M, 410M, 1B, 1.4B, 2.8B) (Biderman et al., 2023), Qwen-2.5 (0.5B, 1.5B) (Yang et al., 2024), Llama-3.2 (1B, 3B) (Grattafiori et al., 2024), and Gemma-3 (270M, 1B) (GemmaTeam et al., 2025).

We report Qwen-2.5 (1.5B) results in Figure 2, results on different model sizes of the same architecture in Appendix 7 and results on different architectures in Appendix 8.

Figure 2a reports mean accessibility for natural-language targets (PG19 spans) of length $n$, using memory lengths $m \in \{1, \ldots, 5\}$. For each fixed $m$, accessibility stays high up to a critical length and then drops sharply. We summarize

each curve by fitting a sigmoid; the fits are consistently good (minimum $R^2 = 0.88$).

Figure 2b shows the same experiment for random token sequences. Despite being out of distribution, we observe the same pattern: beyond a critical length, accessibility falls rapidly with $n$, consistent with the exponential decay predicted by Corollary 4.6 for fixed $m$.

Figure 2c plots the length $n$ at which accessibility crosses 0.5. The scaling is close to linear in both domains ($R^2_{\text{PG19}} = 0.999$, $R^2_{\text{rand.}} = 0.995$), matching the linearity prediction of Corollary 4.6. The slope differs between domains: PG19 targets remain accessible for longer than random strings at the same $m$. A likely reason is that natural text has structure the model can exploit, so the effective amount of information to store is smaller; a similar gap was reported by Kuratov et al. (2025).

In addition, the theoretical slope from Corollary 4.6 is a close upper bound on the slopes estimated from Figure 2c (first row of Table 1), as the factor is between 5 and 10. The small gap is expected: Corollary 4.6 is an upper bound and relies on simplifying geometric assumptions. In particular, it assumes that i) the embedding support is a fully used ball of radius $r$, and ii) each token sequence $\mathbf{t}$ corresponds to a region $\mathcal{E}_{\mathbf{t}}$ of equal volume in the embedding space.

In the following, we relax these worst-case assumptions by better approximating the embedding support $\mathcal{E}$ (Section 5.2) and by empirically estimating the distribution of cell volumes $|\mathcal{E}_{\mathbf{t}}|$ (Section 5.3).

*Table 1.* Ratio between the theoretical upper bound on the slope (using the infinity norm) and the empirical one for natural texts (PG19), across various models. Rows correspond to different enclosure strategies, and columns to the evaluated models. The detailed theoretical analysis is provided in Appendix F.

| | Pythia | | | Qwen-2.5 | | Llama-3.2 | Gemma-3 |
|---|---|---|---|---|---|---|---|
| | 160M | 410M | 1B | 0.5B | 1.5B | 1B | 270M |
| Ball | 9.24 | 9.79 | 7.77 | 14.1 | 20.4 | 14.3 | 11.52 |
| Cone | 9.10 | 9.60 | 7.70 | 14.01 | 20.34 | 13.98 | 11.24 |
| Ellipsoid | 7.92 | 8.15 | 6.12 | 10.96 | 15.30 | 11.86 | 11.12 |
| Ellipsoid + Non-uniform Cells | **6.66** | **5.99** | **4.56** | **7.92** | **10.82** | **10.71** | **8.79** |
| Ellipsoid + variable $\varepsilon$ | 8.65 | 9.83 | 7.71 | 12.32 | 18.81 | 14.63 | 13.42 |

## 5.2. Support Refinement

Prior work shows that transformer embeddings occupy a small and anisotropic subset of the $d$-dimensional ball of radius $r$, rather than filling it uniformly (Rudman et al., 2022). This matters for our bound: if the effective support is smaller than the full ball volume, the accessibility threshold $n^\star(m)$ should be reached at smaller $n$, which lowers the predicted slope.

We upper bound the support by enclosing it within two simple shapes that capture anisotropy: an axis-aligned ellipsoid and a cone. We compute the corresponding packing numbers for each shape and plug them into the slope bound; derivations are deferred to Appendix F. Computing packing numbers requires the radius $r$ for the ball, per-dimension radius $r_i$ for the ellipsoid, and the smallest opening angle that contains all sampled embeddings for the cone. We estimate these support-dependent parameters by Monte Carlo using 10K prompts of length at most $\ell$, generated by sampling tokens i.i.d. from the model vocabulary and concatenating them, and mapping each prompt to an embedding using the same extraction procedure as in the main cramming setup. Experimental details are in Appendix G.4, and the effect of $\ell$ on the resulting upper bounds is shown in Figure 9. In practice, $\ell \approx 1000$ tokens is sufficient to obtain a stable upper-bound estimate.

Using cone or ellipsoid supports, we recompute the theoretical slope and compare it to the empirical slope from Figure 2c. The updated ratios are reported in Table 1 (rows 2 and 3). The gap shrinks for all models, while remaining slightly larger for Qwen-2.5-1.5B.

## 5.3. Cell Volume Distribution

Corollary 4.6 makes an equal-volume approximation: each token (or sequence) corresponds to a region of the embedding space $\mathcal{E}_t$ of equal size. In practice this is unlikely. Common tokens should occupy larger regions under the model's decoder, while rare or specialized tokens may correspond to much smaller regions. Here we measure how uneven these decoder cell volumes are by considering the

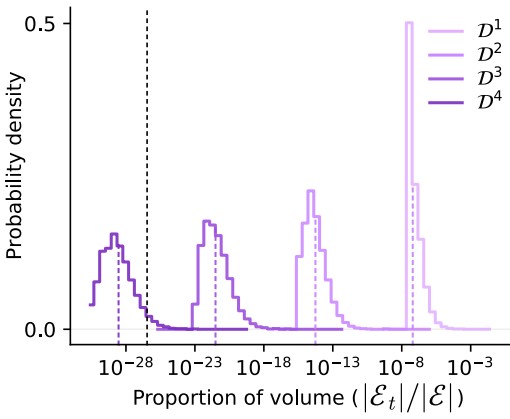

*Figure 3.* Conceptual example of using the cell-volume distribution to tighten the upper bound. Rather than assuming equal-volume cells (Dirac mass), we take the $n$-fold convolution of the empirical one-step volume distribution $\mathcal{D}^1$ (light violet) and track when the median of $\mathcal{D}^n$ (violet dashed) drops below the threshold (black dashed).

empirical distribution $\mathcal{D}$ of the cell volumes of every token $t$ in terms of proportion $\frac{|\mathcal{E}_t|}{|\mathcal{E}|}$.

Concretely, we first approximate the distribution $\mathcal{D}$ corresponding to the next token, then study the $n$-fold multiplicative convolution $\mathcal{D}^n$ to model the distribution of the next $n$ tokens. We compute the smallest $n$ such that the median of $\mathcal{D}^n$ falls below $\frac{1}{\mathcal{P}(\mathcal{E}, \|\cdot\|, \varepsilon)}$, that is when at least half of the sequences become inaccessible (see Figure 3 for a conceptual example). We describe how we obtain the volumes $|\mathcal{E}_t|$ in Appendix H.

*Remark* 5.1. In the uniform setting underlying Corollary 4.6, the distribution $\mathcal{D}$ collapses to a Dirac mass at $\frac{1}{|\mathcal{V}|}$, reflecting equal-volume cells for all tokens. Note that the procedure described above exactly recovers the bound of Corollary 4.6, while nontrivial dispersion in $\mathcal{D}$ yields strictly sharper, data-dependent limits.

Combining this with the ellipsoid support, we recompute the theoretical slope and compare it to the empirical slope from Figure 2c. The updated ratios are reported in Table 1 (row 4). The gap shrinks for all models.

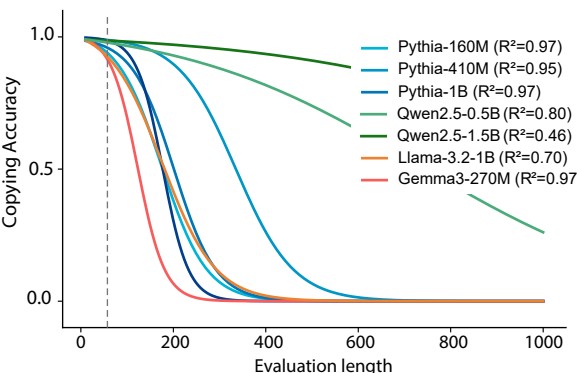

*Figure 4.* Models are trained to copy strings up to a maximum length (grey dashed) and evaluated on generating the exact copy of longer strings. We report exact-match copying accuracy versus string length. For each model, we fit a sigmoid to the accuracy curve (continuous lines) and report the corresponding $R^2$.

*Remark* 5.2. The assumption of uniform precision $\varepsilon$ may be too restrictive. We therefore provide a more refined analysis, whose results are reported in the last row of Table 1. Details are given in Appendix G.2. Although the resulting upper bounds remain valid, they do not account for the fact that most internal representations lie far from 0, which leads to looser bounds in practice.

### 5.4. Copying-Length Generalization.

The first part of this section tested the predictions of Corollary 4.6 using the cramming task. As prompt length $m$ grows, cramming becomes prohibitively expensive, so we evaluate Corollary 4.10 through the copying task instead. Corollary 4.10 predicts a length threshold beyond which most sequences are inaccessible to a fixed transformer $\tau$; it should in particular be impossible for $\tau$ to copy such sequences. Moreover, as for Corollary 4.6, a sharp decline in proportion of accessible sequences is predicted past this threshold.

We test this directly by adding a copying-length generalization experiment following Meyer et al. (2025), avoiding expensive cramming experiments. We fine-tune pretrained models on a synthetic exact string-copying task using next-token cross-entropy loss. Training examples are random strings $x_{1:n}$ of length at most 50, formatted as $x_{1:n} \mid x_{1:n}$, where the model is conditioned on the prefix $x_{1:n} \mid$ and trained to generate the target copy $x_{1:n}$. Training stops at 100% exact-match accuracy on length 50 or after 10K optimization steps We then evaluate exact-match copying accuracy on longer unseen strings within the model context window.

Figure 4 shows a sharp length-dependent transition. Accuracy stays near 100% on short strings, then falls abruptly once string length exceeds a model-specific threshold. Sigmoid fits capture this transition well, with median $R^2 = 0.95$ across models.

These results suggest that the linear regime observed in Figure 2b does not continue indefinitely; every transformer admits an accessibility limitation, independent of the prompt length $m$.

## 6. Conclusion

We established fundamental limits of transformers, showing that accessible sequence length grows at most linearly with prompt size and that most sequences become inaccessible beyond a critical threshold. These results hold even with unbounded context and computation, explaining the empirical failure of transformers to reproduce long sequences. More notably, we theoretically predict the performance of transformers on the cramming task across several architectures, using only the structure of the embedding space.

Importantly, this close characterization of transformers' capacities relies solely on the geometry of the embedding space. This suggests that the embedding space, despite representing only a small fraction of a model's parameters, carries meaningful information about its capabilities.

## 7. Future Work

Besides cramming and copying, another setting closely related to our theoretical results is test-time training, as described in Section 3.1 ("Memory as Context") of Behrouz et al. (2025). The goal of this approach is to mitigate the quadratic cost of attention with respect to context length by compressing a long context into a short sequence of vectors $h$, using a neural network $\mathcal{M}$. In this setting, $\mathcal{M}$ is trained to iteratively encode the context into $h$ via gradient-based updates. Our results provide direct insight into this process: they characterize how the size of $h$ must scale with the length of the context in order to preserve accessibility.

Moreover, as mentioned in Remark 1.1, our results can be extended to other architectures of interest.

## Impact Statement

This paper presents work whose goal is to advance the field of machine learning. There are many potential societal consequences of our work, none of which we feel must be specifically highlighted here.

## Acknowledgements

This research/project is supported by the National Research Foundation, Singapore under its AI Singapore Programme (AISG Award No: AISG3-PhD-2026-01-068T). V. Y. F. Tan is also supported by a Singapore Ministry of Education (MOE) AcRF Tier 2 grant under grant number A-8004062-00-00. This research/project is supported by the National Research Foundation, Singapore under its AI Singapore Programme (AISG Award No: AISG2-PhD-2023-01-040-J), and is part of the programme DesCartes which is supported by the National Research Foundation, Prime Minister's Office, Singapore under its Campus for Research Excellence and Technological Enterprise (CREATE) programme.

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

# A. Transformer Layers

## A.1. Definition

**Definition A.1** (Self-Attention Layer). A $h$-head self-attention layer maps every query token $\mathbf{x}$ within a context $\mathbf{X} \in \mathbb{R}^{d \times m}$ to

$$\text{Att}(\mathbf{x}, \mathbf{X}) = \sum_{i=1}^{h} \mathbf{W}_{\text{o}}^i \mathbf{W}_{\text{v}}^i \mathbf{X} \cdot \sigma\big((\mathbf{W}_{\text{k}}^i \mathbf{X})^{\top} \mathbf{W}_{\text{q}}^i \mathbf{x}\big),$$

where $\mathbf{W}_{\text{q}}^i, \mathbf{W}_{\text{k}}^i \in \mathbb{R}^{s \times d}, \mathbf{W}_{\text{v}}^i \in \mathbb{R}^{s' \times d}, \mathbf{W}_{\text{o}}^i \in \mathbb{R}^{d \times s'}$, $s$ is the hidden dimension (typically $s = s' = \frac{d}{h}$), and the normalization factor $1/\sqrt{s}$ is absorbed into $\mathbf{W}_{\text{k}}^i$. The results for each token are concatenated to produce the output

$$f(\mathbf{X}) := \text{Att}(\mathbf{X}, \mathbf{X})$$
$$= [\text{Att}(\mathbf{X}_{:,1}, \mathbf{X}), \ \dots, \ \text{Att}(\mathbf{X}_{:,m}, \mathbf{X})].$$

**Definition A.2** (Transformer Layers). A transformer layer

$$L_i : \bigsqcup_{m=1}^{+\infty} \mathbb{R}^{d \times m} \longrightarrow \bigsqcup_{m=1}^{+\infty} \mathbb{R}^{d \times m},$$

is defined as

$$\mathbf{X}' = \mathbf{X} + f\big(\text{Norm}(\mathbf{X})\big),$$
$$L_i(\mathbf{X}) = \mathbf{X}' + \text{MLP}_i\big(\text{Norm}(\mathbf{X}')\big),$$

with

$$\text{MLP}(\mathbf{X}) = [\mathbf{W}_2 \text{ReLU}(\mathbf{W}_1 \mathbf{X}_{:,1} + \mathbf{b}_1) + \mathbf{b}_2 + \mathbf{X}_{:,1},$$
$$\dots, \mathbf{W}_2 \text{ReLU}(\mathbf{W}_1 \mathbf{X}_{:,m} + \mathbf{b}_1) + \mathbf{b}_2 + \mathbf{X}_{:,m}],$$

where $\mathbf{W}_1 \in \mathbb{R}^{d_{\text{ff}} \times d}, \mathbf{W}_2 \in \mathbb{R}^{d \times d_{\text{ff}}}, \mathbf{b}_1 \in \mathbb{R}^{d_{\text{ff}}}, \mathbf{b}_2 \in \mathbb{R}^d$, $d_{\text{ff}}$ is the hidden dimension of the MLP and $\text{Norm}$ is a normalization operator applied column-wise such that $\|\text{Norm}(\mathbf{X})\|_\infty$ is bounded for any $\mathbf{X} \in \mathbb{R}^d$. An $l$-layer transformer uses the composition of $l$ transformer layers

$$L = L_l \circ \dots \circ L_1.$$

For simplicity, the positional encoding and masking operation are omitted, following (Kim et al., 2021). Note however that including those operations would not modify any theorem of this paper.

## A.2. Transformer Layers Preserve Bounded Embeddings

For an $l$-layer transformer, a natural upper bound of the output is therefore $\|\mathbf{X}^l\|_\infty \leq (I + kl(A + M))\|E_f\|$, where the maximal initial norm $I = \sup\|\mathbf{X}^0\|_\infty$ is entirely determined by the initial embedding matrix and positional encoding, $N = \|\text{Norm}(\mathbf{X})\|_\infty$, $A = \sup_{\|\mathbf{X}\|_\infty \leq N} \|\text{Attention}(\mathbf{X})\|_\infty$, $M = \sup_{\|\mathbf{X}\|_\infty \leq N} \|\text{MLP}(\mathbf{X})\|_\infty$, and $E_f$ is the final embedding matrix (with the convention $\|\mathbf{X}\|_\infty = \max_i \|\mathbf{X}_i\|$).

We validate this empirically following the same experimental design as in Appendix G.4 in Figure 5.

# B. Mean-Field Transformers

To extend the action of transformers beyond empirical measures to general probability distributions, we introduce the notion of pushforward.

**Definition B.1** (Santambrogio (2015)). Given a probability measure $\mu$ on $\mathbb{R}^d$ and a measurable map $\varphi : \mathbb{R}^d \to \mathbb{R}^d$, the *pushforward* of $\mu$ by $\varphi$, denoted by $\varphi_\sharp \mu$, is the probability measure defined on Borel sets $B \subset \mathbb{R}^d$ by

$$(\varphi_\sharp \mu)(B) := \mu(\varphi^{-1}(B)).$$

Intuitively, $\varphi_\sharp \mu$ is obtained by transporting mass from each point $x$ to its image $\varphi(x)$, preserving total measure. We now define a mean-field transformer, denoting $\mathcal{P}_{\text{c}}(\mathbb{R}^d)$ the set of compactly supported probability measures on $\mathbb{R}^d$.

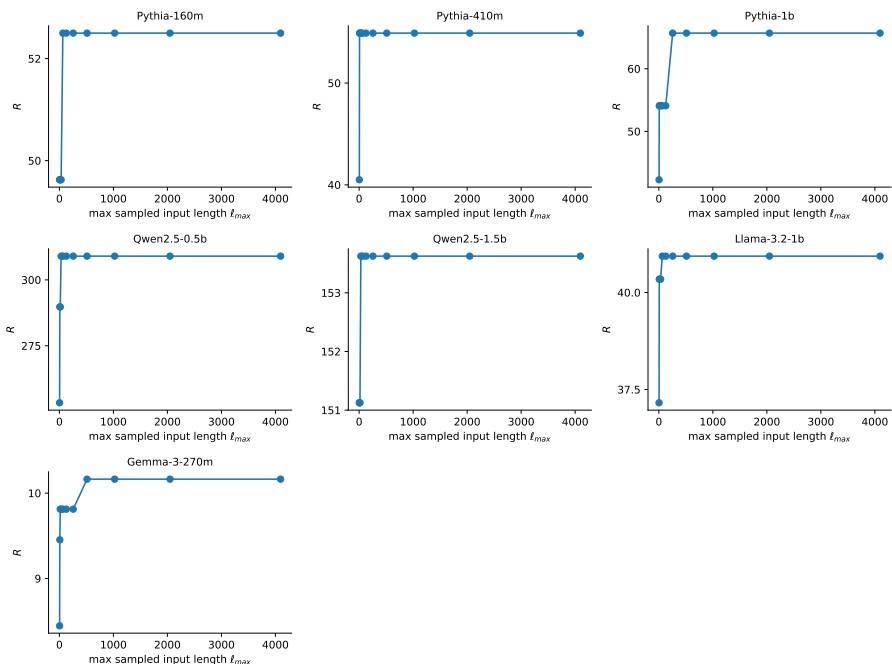

*Figure 5.* Maximal radius $R$ of the internal representations of the transformer for various input lengths.

**Definition B.2** (Mean-Field Self-Attention (Castin et al., 2024)). The mean-field generalization $F$ of any self-attention layer—parameterized by projection matrices $\mathbf{W}_q$, $\mathbf{W}_k$, $\mathbf{W}_v$, and $\mathbf{W}_o$ as defined in Definition A.1—is defined by

$$F : \mu \in \mathcal{P}_c(\mathbb{R}^d) \mapsto (\Gamma_\mu)_\sharp \mu \in \mathcal{P}_c(\mathbb{R}^d), \quad \text{where}$$

$$\Gamma_\mu(\mathbf{x}) := \sum_{i=1}^{h} \frac{\int \mathbf{W}_o^i \mathbf{W}_v^i \mathbf{y} \cdot \exp\left((\mathbf{W}_k^i \mathbf{y})^\top \mathbf{W}_q^i \mathbf{x}\right) \mathrm{d}\mu(\mathbf{y})}{\int \exp\left((\mathbf{W}_k^i \mathbf{y})^\top \mathbf{W}_q^i \mathbf{x}\right) \mathrm{d}\mu(\mathbf{y})}.$$

Mean-field self-attention $F$ generalizes discrete self-attention $\mathrm{Att}$ in the sense that for any input $\mathbf{X} \in \mathbb{R}^{d \times m}$, we have $F(\mathrm{M}(\mathbf{X})) = \mathrm{M}(\mathrm{Att}(\mathbf{X}, \mathbf{X}))$.

**Definition B.3** (Mean-Field Transformer Layer). Similarly, any transformer layer $\tau$, with MLP layer $\mathrm{MLP}$ and mean-field self-attention layer defined by $\Gamma_.$, has the following mean-field generalization $T$.

$$T : \mu \in \mathcal{P}_c(\mathbb{R}^d) \mapsto (\Delta_\mu)_\sharp \mu \in \mathcal{P}_c(\mathbb{R}^d), \quad \text{where}$$

$$\Delta_\mu(\mathbf{x}) := \mathrm{MLP}(\Gamma_\mu(\mathbf{x}) + \mathbf{x}).$$

Similarly to mean-field self-attention, a mean-field transformer layer $T$ generalizes a discrete transformer layer $\tau$ in the sense that for any input $\mathbf{X} \in \mathbb{R}^{d \times m}$, we have $T(\mathrm{M}(\mathbf{X})) = \mathrm{M}(\tau(\mathbf{X}))$ (Meyer et al., 2025). With a slight abuse of notation, we use the same symbol $\tau$ to denote a transformer and its mean-field counterpart.

*Remark* B.4 (Mean-Field Generalization of Masked Attention). We can easily generalize the mean-field framework to masked attention by augmenting the support of the probability measures to $[0, 1] \times \mathbb{R}^d$, as adding $[0, 1]$ allows to encode the position of every token (Castin et al., 2024).

## C. Packing Number

Recall the definition of packing number.

**Definition C.1** (Packing Number). Let $(T, \|\cdot\|)$ be a vector space and let $\varepsilon > 0$.

The $\varepsilon$-*packing number* of $T$, denoted $\mathcal{P}(T, \| \cdot \|, \varepsilon)$, is the maximal number of disjoint open balls of radius $\frac{\varepsilon}{2}$ that can be placed in $T$, or equivalently, the largest cardinality of an $\varepsilon$-separated subset of $T$. That is,

$$\mathcal{P}(T, \| \cdot \|, \varepsilon) = \max\{M \in \mathbb{N} \colon \exists (x_i)_{i \in [M]} \in T^M,$$
$$\forall i \neq j, \|x_i - x_j\| > \varepsilon\}.$$

In this section, we provide upper and lower bounds on the packing number of the sets we are studying.

### THE FINITE PROMPT LENGTH SETTING

We use the following main result to analyze the setting of bounded length prompts.

**Proposition C.2.** *Let $d \in \mathbb{N}$, $r > 0$, and $\varepsilon > 0$. Then*

$$\left(\frac{r}{\varepsilon}\right)^d \leq \mathcal{P}(B^d(0, r), \| \cdot \|, \varepsilon) \leq \left(1 + \frac{2r}{\varepsilon}\right)^d$$

### THE MEAN-FIELD FRAMEWORK

We obtain similar results for the study of prompts of arbitrary length in the mean-field framework. Let $\mathcal{G} := \mathrm{Im}(\mathrm{M}) = \{\mathrm{M}(\mathbf{X}), \mathbf{X} \in \mathbb{R}^{d \times m}, m \in \mathbb{N}\}$ be the set of discrete probability measures on the token embeddings of dimension $d$ as defined in Section 2.2.

**Proposition C.3.** *For any $q \geq 1, \varepsilon > 0$, there exists $C > 0$ such that*

$$\ln\left(\frac{1}{C}\right) + \frac{1}{\varepsilon^d} \leq \ln \mathcal{P}(\mathcal{G}, W_q, 2\varepsilon) \leq \left(1 + \frac{4r}{\varepsilon}\right)^d \ln\left(e + \frac{e\,(2r)^q}{\varepsilon^q}\right),$$

### C.1. Proofs

A central result we need is the volume of the $d$-dimensional ball.

**Lemma C.4** (Wang (2005); Ellis et al. (2007)). *Assume $p_1, \ldots, p_d > 0$. The volume of the generalized unit ball in $\mathbb{R}^d$*

$$B^d_{p_1 p_2 \cdots p_d} = \{\mathbf{x} = (x_1, \ldots, x_d) \ : \ |x_1|^{p_1} + \cdots + |x_d|^{p_d} \leq 1\},$$

*is equal to*

$$\mathrm{Vol}\left(B^d_{p_1 p_2 \cdots p_d}\right) = 2^d \frac{\Gamma\left(1 + \frac{1}{p_1}\right) \cdots \Gamma\left(1 + \frac{1}{p_d}\right)}{\Gamma\left(\frac{1}{p_1} + \frac{1}{p_2} + \cdots + \frac{1}{p_d} + 1\right)}.$$

*Note that we can take some of the $p_i$'s to be $+\infty$ with the convention $\frac{1}{+\infty} = 0$.*

*In particular, the volume of the $l_p$ ball of radius $r$ is*

$$\mathrm{Vol}\left(B^d_p(0, r)\right) = (2r)^d \frac{\Gamma\left(\frac{p+1}{p}\right)^d}{\Gamma\left(\frac{p+d}{p}\right)}.$$

*And the one corresponding to the $l_\infty$ norm is*

$$\mathrm{Vol}\left(B^d_\infty(0, r)\right) = (2r)^d.$$

Proposition C.2 can be proved using known bounds for the packing number of a general bounded set $K$, depending on the volume of $K$.

**Lemma C.5** (Vershynin (2026, Proposition 4.2.10)). *Let $d \in \mathbb{N}$, $K \subset \mathbb{R}^d$ and $\varepsilon > 0$. Then,*

$$\frac{|K|}{|\varepsilon B^d(0, 1)|} \ \leq \ \mathcal{P}(K, \| \cdot \|, \varepsilon), \quad and$$

$$\mathcal{P}(K, \| \cdot \|, \varepsilon) \ \leq \ \frac{|K + (\varepsilon/2)B^d(0, 1)|}{|(\varepsilon/2)B^d(0, 1)|}.$$

We can then deduce Proposition C.3 from the following two lemmas.

**Lemma C.6** (Nguyen (2013, Lemma 4.b)). *For any $q \geq 1, \varepsilon > 0$,*

$$\ln \mathcal{P}(\mathcal{G}, W_q, 4\varepsilon) \leq \mathcal{P}(\Theta, \| \cdot \|_q, \varepsilon) \ln \left( e + \frac{e \operatorname{Diam}(\Theta)^q}{\varepsilon^q} \right),$$

*with $\Theta = B^d(0, r)$ and $e = \exp(1)$.*

**Lemma C.7** (Meyer et al. (2025, Proposition 3.6)). *For any $q \geq 1, \varepsilon > 0$, there exists $C > 0$ such that*

$$\mathcal{P}(\mathcal{G}, W_q, \varepsilon) \geq \frac{1}{C} \exp(\frac{1}{\varepsilon^d}).$$

## D. Permutation-Invariant Norms and Wasserstein Distances

In this section, we discuss the Wasserstein distance, and specifically how the Wasserstein distance between two empirical measures $\mathrm{M}(\mathbf{X})$ and $\mathrm{M}(\mathbf{Y})$ relates to the classical distance $\|\mathbf{X} - \mathbf{Y}\|$. This allows us to study the validity of Assumption 4.8.

**Definition D.1** (Wasserstein Distance (Santambrogio, 2015)). For $q \geq 1$, the $\ell_q$ Wasserstein distance on $\mathcal{P}_{\mathrm{c}}(\mathbb{R}^d)$ is given by

$$W_q(\mu, \nu) = \left( \inf_{\pi \in \Pi(\mu, \nu)} \int \|x - y\|^q \, d\pi(x, y) \right)^{1/q},$$

for $\mu, \nu \in \mathcal{P}_{\mathrm{c}}(\mathbb{R}^d)$, where $\Pi(\mu, \nu)$ is the set of couplings between $\mu$ and $\nu$, *i.e.* of probability measures $\pi \in \mathcal{P}(\mathbb{R}^d \times \mathbb{R}^d)$ such that $\int \pi(\cdot, y) \, dy = \mu$ and $\int \pi(x, \cdot) \, dx = \nu$.

The $\ell_\infty$ Wasserstein distance is defined as the limit of the $W_q$ distance as $q$ goes to infinity.

$$W_\infty(\mu, \nu) = \lim_{q \to \infty} W_q(\mu, \nu)$$

In particular, two sequences $\mathbf{X}, \mathbf{Y} \in \mathbb{R}^{d \times n}$ of equal length $n$ are of Wasserstein distance

$$W_q(\mathrm{M}(\mathbf{X}), \mathrm{M}(\mathbf{Y})) = \left( \min_{\pi \in S_n} \frac{1}{n} \sum_{i=1}^{n} \|X_{:,i} - Y_{:,\pi(i)}\|^q \right)^{1/q},$$

$$W_\infty(\mathrm{M}(\mathbf{X}), \mathrm{M}(\mathbf{Y})) = \min_{\pi \in S_n} \max_{i=1,\ldots,n} \|X_{:,i} - Y_{:,\pi(i)}\|.$$

The permutation invariance of the mean-field transformer architecture prompts us to consider permutation invariant versions of the $\ell_p$ and $\ell_\infty$ distances.

**Definition D.2.** Consider two sequences $\mathbf{X}, \mathbf{Y} \in \mathbb{R}^{d \times n}$ of equal length $n$. For $\pi \in S_n$, let $\mathbf{Y}_\pi = (\mathbf{Y}_{:,\pi(1)}, \ldots, \mathbf{Y}_{:,\pi(n)})$. Then

$$d_q(\mathbf{X}, \mathbf{Y}) = \min_{\pi \in S_n} \left\| \mathbf{X} - \mathbf{Y}_\pi \right\|_q, \quad 1 \leq q < \infty,$$

$$d_\infty(\mathbf{X}, \mathbf{Y}) = \min_{\pi \in S_n} \left\| \mathbf{X} - \mathbf{Y}_\pi \right\|_\infty,$$

where $S_n$ denotes the set of permutations of $[n]$.

We can now relate Wasserstein distance between empirical measures and classical distance.

**Proposition D.3.** *Consider two sequences $\mathbf{X}, \mathbf{Y} \in \mathbb{R}^{d \times n}$ of equal length $n$. Then,*

$$W_q(\mathrm{M}(\mathbf{X}), \mathrm{M}(\mathbf{Y})) = n^{-1/q} d_q(\mathbf{X}, \mathbf{Y}), \quad 1 \leq q < \infty,$$

$$W_\infty(\mathrm{M}(\mathbf{X}), \mathrm{M}(\mathbf{Y})) = d_\infty(\mathbf{X}, \mathbf{Y}).$$

Notice that bounding the Wasserstein distance between two empirical measures $M(\mathbf{X})$ and $M(\mathbf{Y})$ does not directly bound the distance between $\mathbf{X}$ and $\mathbf{Y}$. The latter bound depends on the lengths of both vectors. Therefore, a finite precision on vectors does not directly imply a finite Wasserstein precision (that is Assumption 4.3 does not directly imply Assumption 4.8).

However, one can clearly see that Assumption 4.3 is very weak for large vectors. Indeed, if a transformer cannot distinguish two vectors whose $\ell_1$-distance is at most $\varepsilon$ when their length is 1, then for vectors of length $n$, this tolerance scales roughly as $n\varepsilon$. In other words, the same absolute $\ell_1$-precision becomes increasingly coarse as the vector length grows: small perturbations distributed across many coordinates can yield an $\ell_1$ deviation of order $n\varepsilon$ while remaining imperceptible to the model.

Of course, assuming a uniform $\varepsilon$-precision in the $\ell_1$ norm is overly restrictive. Yet, the argument above suggests that for some $q \geq 1$, the effective distinguishability of the model may scale as $n^{-1/q}\varepsilon$. Therefore, Proposition D.3 provides theoretical support for Assumption 4.8: a finite vector precision naturally induces a finite Wasserstein precision when the appropriate scaling with $n$ and $q$ is taken into account.

# E. Arbitrary Prompt Length via Elementary Operations

Contrary to Assumption 4.3, Assumption 4.8 is non-trivial. We show in this section that it can be replaced by a much weaker assumption, pertaining to what we call elementary operations.

**Definition E.1** (Elementary Operations on a Sequence of Vectors). Let $\mathbf{X} = [\mathbf{x}_1, \ldots, \mathbf{x}_n]$ be a sequence of $n$ vectors. An elementary operation on $\mathbf{X}$ consists in duplicating or removing one of its tokens.

**Assumption E.2** (Elementary Operation Precision of a Transformer). Let $\tau$ be a transformer. There exists a constant $p \in \mathbb{N}$ such that, if a sequence of tokens $\mathbf{Y}$ can be recovered from another sequence $\mathbf{X} \in \mathbb{R}^{d \times n}$ of length $n$ through at most $\frac{n}{p}$ elementary operations, then $\tau$ cannot distinguish between $\mathbf{Y}$ and $\mathbf{X}$. We say that $\frac{1}{p}$ is the elementary operation precision (e.o. precision) of $\tau$.

This assumption states that a transformer $\tau$ is insensitive to small structural edits of its input sequence. The only allowed operations are token deletions and duplications—no arbitrary modifications of token content. It is therefore reasonable to expect that if $\mathbf{Y}$ differs from $\mathbf{X}$ by only a small fraction of such operations, then $\tau$ produces nearly identical representations. Although we do not specify the exact constant $p$, the claim clearly fails for large perturbations (e.g., modifying half the tokens) and is trivially true for extremely small ones (e.g., $0.001\%$ of the tokens). Hence, it is highly plausible that there exists a non-trivial fraction $1/p$ for which Assumption E.2 holds.

Under Assumption E.2, we can obtain an upper bound on the number of different inputs a given transformer $\tau$ can distinguish between, and hence on the amount of distinct output sequences it can access. This upper bound is a function of both the e.o. precision $p$ of the transformer, and the number of distinct input values $D$ it can distinguish between. If we consider a hard prompt transformer [Definition 2.3] without positional embedding, then $D = |\mathcal{V}|$. In general, we can take $D = (\frac{r}{\varepsilon})^d$, where $\varepsilon$ is the standard precision of $\tau$ and $r$ is its embedding radius.

**Theorem E.3.** *There exists an upper bound $f(p, D)$ on the number of sequences of tokens that are accessible by a transformer $\tau$ with elementary operation precision $\frac{1}{p}$.*

*Proof.* We showed that the input space of $\tau$ is exactly the set of empirical distributions over $[D]$, which is a subset of $\mathbb{N}^D$. That is the transformer maps to the same image two inputs $\mathbf{x}$ and $\mathbf{y}$ satisfying $\exists q \in \mathbb{Q}, \mathbf{x} = q\mathbf{y}$. We denote by $\mathcal{R}$ and $\bar{\mathbf{x}}$ the associated equivalence relation and equivalence classes:

- $\forall \mathbf{x}, \mathbf{y} \in \mathbb{N}^D, \mathbf{x}\mathcal{R}\mathbf{y} \iff \exists q \in \mathbb{Q}, \mathbf{x} = q\mathbf{y}$.

- $\forall \mathbf{x} \in \mathbb{N}^D, \bar{\mathbf{x}} = \{\mathbf{y} \in \mathbb{N}^D : \mathbf{x}\mathcal{R}\mathbf{y}\}$.

Then the input space of $\tau$ is exactly $\mathbb{N}^D/\mathcal{R} = \{\bar{\mathbf{x}} : \mathbf{x} \in \mathbb{N}^D\}$. An important thing to note is that in $\mathbb{N}^D/\mathcal{R}$, the length of an empirical distribution $\mathbf{x}$ is simply $\|\mathbf{x}\|_1$, and an elementary operation on $\mathbf{x}$ amounts to adding or subtracting 1 to a non-zero coefficient of $\mathbf{x}$.

Take the basis $\mathcal{B} = \{\mathbf{x} \in \mathbb{N}^D : \|\mathbf{x}\|_\infty < b\}$. Theorem E.3 is a direct consequence of the fact that every vector $\bar{\mathbf{x}} \in \mathbb{N}^D/\mathcal{R}$ is at most $g(b, D)\|\mathbf{x}\|_1$ elementary operations away from a basis vector for some function $g$. Then, we conclude the proof by choosing $b$ such that $g(b, D) \leq \frac{1}{p}$, and obtain $f(p, D) = |\mathcal{B}| = b^D$.

A very straightforward way of proving this result is presented in this section. We obtain $g(b, D) = \frac{D}{2(b-2)}$ and $b = \left\lceil \frac{pD}{2} \right\rceil$ in Lemma E.4. We provide in Appendix E more complicated proofs leading to tighter bounds. □

**Lemma E.4.** *Let $\mathbf{x} \in \mathbb{N}^D$. There exists $\bar{\mathbf{y}} \in \bar{\mathcal{B}} := \{\bar{\mathbf{y}} \colon \mathbf{y} \in \mathcal{B}\}$ such that $\|\mathbf{x} - \mathbf{y}\|_1 \le \frac{D\|\mathbf{x}\|_1}{2(b-2)}$.*

*Proof.* Let $\mathbf{x} = (x_1, \ldots, x_D) \in \mathbb{N}^D$. Without loss of generality, assume that $x_1 \le \ldots \le x_D$. Let $l \in \mathbb{N}$ be the quotient of the euclidean division of $x_D$ by $b - 2$: $l(b-2) \le x_D < l(b-1)$. Then there exists $\mathbf{y} = (y_1, \ldots, y_D) \in \mathbb{N}^D$ such that $\|\mathbf{x} - \mathbf{y}\|_1 \le D\lfloor \frac{l}{2} \rfloor$ and $\forall i \in [D], l | y_i$ (every $y_i$ is the closest multiple of $l$ to $x_i$).

Clearly, $\frac{1}{l}\mathbf{y} \in \mathcal{B}$ and $\mathbf{y}\mathcal{R}(\frac{1}{l}\mathbf{y})$. Hence $\bar{\mathbf{y}} = \frac{1}{l}\bar{\mathbf{y}} \in \bar{\mathcal{B}}$.

Finally, $l \le \frac{x_D}{b-2} \le \frac{\|\mathbf{x}\|_1}{b-2}$ so $\|\mathbf{x} - \mathbf{y}\|_1 \le \frac{D\|\mathbf{x}\|_1}{2(b-2)}$. □

### E.1. Tighter Bounds for Theorem E.3

**Theorem E.3.** *There exists an upper bound $f(p, D)$ on the number of sequences of tokens that are accessible by a transformer $\tau$ with elementary operation precision $\frac{1}{p}$.*

Recall that to prove Theorem E.3, for every $p \in \mathbb{N}$ we obtain a basis $\mathcal{B} = \{\mathbf{x} \in \mathbb{N}^D \colon \|\mathbf{x}\|_\infty < b\}$ for some $b \in \mathbb{N}$ that is $\frac{1}{p}$-dense in $\mathbb{N}^D$ in the sense that

$$\forall \mathbf{x} \in \mathbb{N}^D, \exists \mathbf{y} \in \mathbb{N}^D, \begin{cases} \bar{\mathbf{y}} \in \bar{\mathcal{B}}, \\ \|\mathbf{x} - \mathbf{y}\|_1 \le \frac{\|\mathbf{x}\|_1}{p}. \end{cases}$$

Then, clearly $f(p, D) \le |\mathcal{B}| = b^D$.

*Remark* E.5. Note that the last equality $f(p, D) \le |\mathcal{B}| = b^D$ leaves room for improvement as $f(p, D) \le |\bar{\mathcal{B}}| < |\mathcal{B}| = b^D$.

In Lemma E.4, we propose $b = \left\lceil \frac{pD}{2} \right\rceil$ for the basis radius. Let us show that $b = 2p$ suffices.

**Lemma E.6.** *Let $\mathbf{x} \in \mathbb{N}^D$. There exists $\bar{\mathbf{y}} \in \bar{\mathcal{B}} := \{\bar{\mathbf{y}} \colon \mathbf{y} \in \mathcal{B}\}$ such that $\|\mathbf{x} - \mathbf{y}\|_1 \le \frac{2\|\mathbf{x}\|_1}{b}$.*

*Proof.* For the purpose of this proof, we introduce the distance d on $\mathbb{N}^D/\mathcal{R}$ defined as $d(\bar{\mathbf{x}}, \bar{\mathbf{y}}) = \min_{\mathbf{x} \in \bar{\mathbf{x}}, \mathbf{y} \in \bar{\mathbf{y}}} \|\mathbf{x} - \mathbf{y}\|_1$.

Let $\mathbf{x} = (x_1, \ldots, x_D) \in \mathbb{N}^D$. By considering separately the cases where $\min_{i \in [D]} x_i \le \frac{\|\mathbf{x}\|_1}{b^2}$ and $\min_{i \in [D]} x_i > \frac{\|\mathbf{x}\|_1}{b^2}$, we can show that there exists $\mathbf{y}^1 = (0, \ldots, 0, y, n_1 y, \ldots, n_k y) \in \mathbb{N}^D$ such that $d(\bar{\mathbf{x}}, \overline{\mathbf{y}^1}) \le \frac{\|\mathbf{x}\|_1}{b}$.

Without loss of generality, assume that $n_1 \le \ldots \le n_k$. If $n_k \le b$, then $\overline{\mathbf{y}^1} = \overline{(0, \ldots, 0, 1, n_1, \ldots, n_k)} \in \bar{\mathcal{B}}$. Else, consider $\mathbf{y}^2 = (0, \ldots, 0, 0, n_1 y, \ldots, n_k y) \in \mathbb{N}^D$ which satisfies $\|\mathbf{y}^1 - \mathbf{y}^2\|_1 = y = \frac{\|\mathbf{y}^1\|_1}{1+n_1+\cdots+n_k} < \frac{\|\mathbf{y}^1\|_1}{1+kb}$.

We can conclude by induction. □

## F. Theoretical Slope of the Cramming Task for Various Supports

We describe how we obtain the formulas for the various slopes in the following subsections. First, let us introduce a few useful notations and equations.

Let $\mathcal{D}$ be the distribution over the volume proportion $\frac{|\mathcal{E}_t|}{|\mathcal{E}|}$ of the next-token cells. That is $\mathcal{D}(I) = \frac{1}{|\mathcal{V}|} \sum_{t \in \mathcal{V}} \mathbf{1}_{\frac{|\mathcal{E}_t|}{|\mathcal{E}|} \in I}$ for all $I \subset \mathbb{R}^+$. We can approximate the distribution over the volume fractions $\frac{|\mathcal{E}_t|}{|\mathcal{E}|}$ of the sequences of length $n$, $\mathbf{t} \in \mathcal{V}^n$ by

$$\begin{aligned} \mathcal{D}_n(I) &= \frac{1}{|\mathcal{V}|^n} \sum_{\mathbf{t} \in \mathcal{V}^n} \mathbf{1}_{\frac{|\mathcal{E}_\mathbf{t}|}{|\mathcal{E}|} \in I} \\ &\simeq \frac{1}{|\mathcal{V}|^n} \sum_{\mathbf{t} \in \mathcal{V}^n} \mathbf{1}_{(\prod_{i=1}^n \frac{|\mathcal{E}_{\mathbf{t}_i}|}{|\mathcal{E}|}) \in I} \\ &= \mathcal{D}^n(I), \end{aligned} \tag{3}$$

where $\mathcal{D}^n$ denotes the distribution of the product of $n$ variables $X_1, ..., X_n \overset{\text{iid}}{\sim} \mathcal{D}$. Then, at least half of the token sequences of length $n$ are inaccessible if $\text{Med}(\mathcal{D}^n) \leq \frac{1}{\mathcal{P}(\mathcal{E}, \|\cdot\|, \varepsilon)}$, where $\text{Med}(\mathcal{D}^n)$ denotes the median of $\mathcal{D}^n$. Note that in the setting of Corollary 4.6, $\mathcal{D}$ is equal to the Dirac distribution $\delta_{\frac{1}{|\mathcal{V}|}}$, and we recover the slope $\frac{\ln(\mathcal{P}(\mathcal{E}, \|\cdot\|, \varepsilon))}{\ln(|\mathcal{V}|)}$ from

$$\text{Med}(\mathcal{D}^n) \leq \frac{1}{\mathcal{P}(\mathcal{E}, \|\cdot\|, \varepsilon)} \quad \Longleftrightarrow \quad \frac{1}{|\mathcal{V}|^n} \leq \frac{1}{\mathcal{P}(\mathcal{E}, \|\cdot\|, \varepsilon)}$$

$$\Longleftrightarrow \quad n \geq \frac{\ln(\mathcal{P}(\mathcal{E}, \|\cdot\|, \varepsilon))}{\ln(|\mathcal{V}|)}. \tag{4}$$

*Remark* F.1. In Eq. 3, we assume conditional independence of next-token cells: for any realized prefix, the distribution of subsequent cell volumes within the selected cell is again $\mathcal{D}$. Although strong, this assumption is conservative: we hypothesize that adding context reduces the number of plausible next tokens, making the next-token distribution sharper than the marginal and thus yielding more variance in the volume cells than $\mathcal{D}$. Under this view, the i.i.d. model $\mathcal{D}^n$ is expected to overestimate typical continuation volumes and can be interpreted as an upper bound.

### F.1. When the Support Is a Ball

**Proposition F.2.** *Let $B^{d \times m}(0, r) \supset \mathcal{E}$ be an enclosure of the embedding support. Then the slope of the Cramming Task satisfies $C \leq d \frac{\ln(1 + \frac{2r}{\varepsilon})}{\ln(|\mathcal{V}|)}$.*

*Proof.* This is a simple corollary of Equation 4 and Proposition C.2. $\qquad\square$

### F.2. When the Support Is a Cone

Let $n \geq 2$, $r > 0$, and $\delta \in (0, \pi)$. Define the (infinite) cone of angle $\delta$

$$C_\delta := \left\{ \mathbf{x} \in \mathbb{R}^n : \mathbf{x}_1 \tan(\delta/2) > \sqrt{\sum_{i=2}^n \mathbf{x}_i^2} \right\},$$

and its truncation by the Euclidean ball $B^d(0, r) := \{\mathbf{x} \in \mathbb{R}^d : \sqrt{\sum_{i=1}^d \mathbf{x}_i^2} < r\}$:

$$C_{\delta, r} := C_\delta \cap B^d(0, r).$$

**Proposition F.3.** *Let $C_{\delta, r}^m \supset \mathcal{E}$ be an enclosure of the embedding support. Then the slope of the Cramming Task satisfies*

$$C \leq \frac{1}{\ln(|\mathcal{V}|)} \ln\left( \left(1 + \frac{2r}{\varepsilon}\right)^d \frac{\Gamma(\frac{d+2}{2})}{2r\Gamma(\frac{3}{2})\Gamma(\frac{d+1}{2})} \left[ \frac{1}{d} \sin^{d-1}\left(\frac{\delta}{2}\right) \cos\left(\frac{\delta}{2}\right) + \frac{1}{2}\left( B\left(\frac{1}{2}, \frac{d+1}{2}\right) - B\left(\cos^2\left(\frac{\delta}{2}\right); \frac{1}{2}, \frac{d+1}{2}\right)\right) \right] \right),$$

*where $B(a, b) = \int_0^1 t^{a-1}(1-t)^{b-1}\, dt$ is the Beta function, and $B(x; a, b) = \int_0^x t^{a-1}(1-t)^{b-1}\, dt$ is the incomplete Beta function.*

*Proof.*

**Lemma F.4.** *The volume of the $d$-dimensional cone of angle $\delta$ and radius $r$ satisfies*

$$|C_{\delta, r}| = \omega_{d-1} r^d \left[ \frac{1}{d} \sin^{d-1}\left(\frac{\delta}{2}\right) \cos\left(\frac{\delta}{2}\right) + \frac{1}{2}\left( B\left(\frac{1}{2}, \frac{d+1}{2}\right) - B\left(\cos^2\left(\frac{\delta}{2}\right); \frac{1}{2}, \frac{d+1}{2}\right)\right) \right],$$

*where $\omega_{d-1} := |B^{d-1}(0, 1)|$ is the volume of the unit ball in $\mathbb{R}^{d-1}$.*

*Proof of Lemma F.4.* **Step 1: Scaling Reduction.** For any $r > 0$, the change of variables $\mathbf{x} = r\mathbf{y}$ gives

$$|C_\delta \cap B^d(0, r)| = \int_{\mathbb{R}^d} \mathbf{1}_{C_\delta}(\mathbf{x}) \mathbf{1}_{B^d(0, r)}(\mathbf{x})\, d\mathbf{x} = \int_{\mathbb{R}^d} \mathbf{1}_{C_\delta}(r\mathbf{y}) \mathbf{1}_{B^d(0,1)}(\mathbf{y})\, r^d\, d\mathbf{y} = r^d |C_\delta \cap B^d(0, 1)|,$$

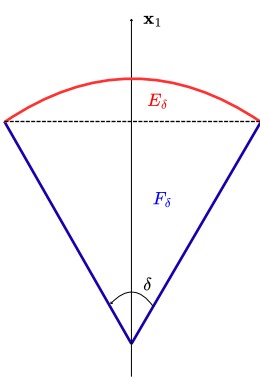

*Figure 6.* Decomposition of the cone in $E_\delta$ and $F_\delta$.

since $C_\delta$ is a cone (i.e. $\mathbf{y} \in C_\delta \iff r\mathbf{y} \in C_\delta$ for $r > 0$). Thus it suffices to compute $|C_\delta \cap B^d(0, 1)|$.

**Step 2: Disjoint Decomposition.** Write, as shown in Figure 6,

$$E_\delta := (C_\delta \cap B^d(0,1)) \cap \{0 < \mathbf{x}_1 < \cos(\delta/2)\}, \qquad F_\delta := (C_\delta \cap B^d(0,1)) \cap \{\cos(\delta/2) < \mathbf{x}_1 < 1\}.$$

Then $C_\delta \cap B^d(0,1) = E_\delta \ \dot\cup\ F_\delta$ up to a null set (the slice $\mathbf{x}_1 = \cos(\delta/2)$), so

$$|C_\delta \cap B^d(0,1)| = |E_\delta| + |F_\delta|.$$

**Step 3: Volume of the Conical Part $E_\delta$.** Fix $\rho \in (0, \cos(\delta/2))$. On the hyperplane $\{\mathbf{x}_1 = \rho\}$, we have

$$\sqrt{\sum_{i=2}^{d} \mathbf{x}_i^2} < \mathbf{x}_1 \tan(\delta/2) = \rho \tan(\delta/2).$$

Since $\rho < \cos(\delta/2)$, we obtain

$$\rho \tan(\delta/2) \leq \cos(\delta/2) \tan(\delta/2) = \sin(\delta/2).$$

Hence $\sum_{i=1}^{d} \mathbf{x}_i^2 = \rho^2 + \sum_{i=2}^{d} \mathbf{x}_i^2 < \rho^2 + \sin^2(\delta/2) \leq 1.$

So the ball constraint $\sqrt{\sum_{i=1}^{d} \mathbf{x}_i^2} < 1$ is inactive on these slices and

$$\{\mathbf{x}_1 = \rho\} \cap E_\delta = \{\rho\} \times B^{d-1}(0, \rho \tan(\delta/2)).$$

Thus,

$$|E_\delta| = \int_0^{\cos(\delta/2)} |B^{d-1}(0, \rho \tan(\delta/2))| \, d\rho = \int_0^{\cos(\delta/2)} \omega_{d-1} \big(\rho \tan(\delta/2)\big)^{d-1} \, d\rho.$$

Evaluating the integral yields

$$|E_\delta| = \omega_{d-1} \tan^{d-1}(\delta/2) \int_0^{\cos(\delta/2)} \rho^{d-1} \, d\rho = \frac{\omega_{d-1}}{d} \tan^{d-1}(\delta/2) \cos^d(\delta/2) = \frac{\omega_{d-1}}{d} \sin^{d-1}(\delta/2) \cos(\delta/2).$$

**Step 4: Volume of the Spherical-Cap Part $F_\delta$.** Fix $\rho \in (\cos(\delta/2), 1)$. On the hyperplane $\{\mathbf{x}_1 = \rho\}$, the ball constraint $\sqrt{\sum_{i=1}^{d} \mathbf{x}_i^2} < 1$ is equivalent to $\sqrt{\sum_{i=2}^{d} \mathbf{x}_i^2} < \sqrt{1 - \rho^2}$. For $\rho > \cos(\delta/2)$,

$$\sqrt{1 - \rho^2} < \sqrt{1 - \cos^2(\delta/2)} = \sin(\delta/2) = \cos(\delta/2) \tan(\delta/2) < \rho \tan(\delta/2),$$

so the cone constraint is inactive on these slices: the ball provides the tighter bound. Thus

$$\{\mathbf{x}_1 = \rho\} \cap F_\delta = \{\rho\} \times B^{d-1}(0, \sqrt{1 - \rho^2}).$$

Hence,

$$|F_\delta| = \int_{\cos(\delta/2)}^1 |B^{d-1}(0, \sqrt{1-\rho^2})| \, d\rho = \omega_{d-1} \int_{\cos(\delta/2)}^1 \left(1 - \rho^2\right)^{\frac{d-1}{2}} d\rho.$$

Let $\tau = \rho^2$, so $d\rho = \frac{1}{2}\tau^{-1/2} \, d\tau$. Then

$$|F_\delta| = \frac{\omega_{d-1}}{2} \int_{\cos^2(\delta/2)}^1 \tau^{-1/2}(1-\tau)^{\frac{d-1}{2}} \, d\tau = \frac{\omega_{d-1}}{2} \left[ B\left(\tfrac{1}{2}, \tfrac{d+1}{2}\right) - B\left(\cos^2(\tfrac{\delta}{2}); \tfrac{1}{2}, \tfrac{d+1}{2}\right) \right].$$

**Step 5: Conclusion.** Summing the two contributions,

$$|C_\delta \cap B^d(0,1)| = \omega_{d-1} \left[ \frac{1}{d} \sin^{d-1}\left(\frac{\delta}{2}\right) \cos\left(\frac{\delta}{2}\right) + \frac{1}{2}\left( B\left(\tfrac{1}{2}, \tfrac{d+1}{2}\right) - B\left(\cos^2(\tfrac{\delta}{2}); \tfrac{1}{2}, \tfrac{d+1}{2}\right) \right) \right],$$

and scaling gives the claimed formula for $|C_\delta \cap B^d(0,r)| = r^d |C_\delta \cap B^d(0,1)|$. $\qquad\square$

The packing number of $C_{\delta,r}$ follows by symmetry (or using Lemma C.5).

**Corollary F.5.** *The packing number of the d-dimensional cone of angle $\delta$ and radius $r$ satisfies*

$$\mathcal{P}(C_{\delta,r}, \|\cdot\|_2, \varepsilon) \lesssim \frac{\Gamma(\frac{d+2}{2})}{2r\Gamma(\frac{3}{2})\Gamma(\frac{d+1}{2})} \left[ \frac{1}{d} \sin^{d-1}\left(\frac{\delta}{2}\right) \cos\left(\frac{\delta}{2}\right) + \frac{1}{2}\left( B\left(\tfrac{1}{2}, \tfrac{d+1}{2}\right) - B\left(\cos^2(\tfrac{\delta}{2}); \tfrac{1}{2}, \tfrac{d+1}{2}\right) \right) \right] \mathcal{P}(B^d(0,r), \|\cdot\|_2, \varepsilon)$$

Finally, we can conclude using Equation 4 and Proposition C.2.

$\qquad\square$

### F.3. When the Support Is an Ellipsoid

We use here the $l_\infty$ norm.

**Proposition F.6.** *Let* $\left( [x_1^{\min}, x_1^{\max}] \times \cdots \times [x_d^{\min}, x_d^{\max}] \right)^m \supset \mathcal{E}$ *be an enclosure of the embedding support. Then the slope of the Cramming Task satisfies* $C \le \frac{\sum_{i=1}^d \ln(1 + \frac{x_i^{\max} - x_i^{\min}}{\varepsilon})}{\ln(|\mathcal{V}|)}$.

*Proof.* This is a simple corollary of Equation 4 and Proposition C.2. $\qquad\square$

## G. Supplementary Experimental Results

### G.1. Latent space constants for each model

| | Pythia | | | Qwen-2.5 | | Llama-3.2 | Gemma-3 |
|---|---|---|---|---|---|---|---|
| | 160M | 410M | 1B | 0.5B | 1.5B | 1B | 270M |
| Hidden dimension $d$ | 768 | 1024 | 2048 | 896 | 1536 | 2048 | 640 |
| Vocabulary Size $|\mathcal{V}|$ | 50304 | 50304 | 50304 | 151936 | 151936 | 128256 | 262144 |
| Radius $r$ | 63.62 | 54.90 | 65.68 | 309.00 | 153.62 | 41.06 | 10.74 |
| Cone angle $\theta$ | 1.96 | 1.87 | 2.22 | 2.23 | 2.34 | 1.73 | 1.82 |

*Table 2.* Model constants used to estimate slope upper-bound ratios in Table 1. The dimensionality $d$ and vocabulary size $|\mathcal{V}|$ are determined by the model architecture, while $r$ and $\theta$ are estimated in Section 5.2.

### G.2. Estimation of precision $\varepsilon$

#### G.2.1. FIXED PRECISION

We assume a finite numerical precision in transformer computations: inputs closer than a threshold $\varepsilon$ are effectively indiscernible. In practice, transformers are typically evaluated using half-precision floating-point arithmetic (fp16), which uses $k_{\max} = 5$ exponent bits and $p = 11$ bits of significand precision.[2] For numbers with exponent $k$, the spacing between representable values is $2^{k-p}$. Thus, two values at exponent $k$ become indistinguishable after rounding if their difference is below this spacing.

We use the standard interval definition of machine epsilon: the gap between 1 and the next representable floating-point value, i.e. $\varepsilon = 2^{-10}$, as used by Pytorch (Paszke et al., 2019). This definition lets us instantiate precision-dependent constants for fp16 and derive upper bounds on model slopes.

#### G.2.2. VARIABLE PRECISION

One could however argue that this definition is not precise: the precision should vary depending on the order of magnitude of the number being represented. To that end, we revise the upper bound of the packing number as follows.

Let $\mathcal{E} = [r_{\min}, r_{\max}]$ be the embedding space, where $0 < r_{\min} \leq r_{\max}$ (by symmetry, we can easily extend to the setting where $\mathcal{E} = [-r_{\max}^-, -r_{\min}^-] \cup [r_{\min}^+, r_{\max}^+]$).

Define
$$a = \lceil \log_2 r_{\min} \rceil, \qquad b = \lceil \log_2 r_{\max} \rceil.$$

Then
$$2^{a-1} < r_{\min} \leq 2^a, \qquad 2^{b-1} < r_{\max} \leq 2^b.$$

And
$$[r_{\min}, r_{\max}] = [r_{\min}, 2^a] \cup [2^a, 2^{a+1}] \cup \cdots \cup [2^{b-1}, r_{\max}].$$

Hence, by subadditivity of the packing number,

$$\mathcal{P}(\mathcal{E}) \leq \mathcal{P}([r_{\min}, 2^a]) + \sum_{j=a}^{b-2} \mathcal{P}([2^j, 2^{j+1}]) + \mathcal{P}([2^{b-1}, r_{\max}]).$$

Using the precision at scale $2^j$, $\varepsilon_j = 2^{j-12}$, we obtain - $\mathcal{P}([r_{\min}, 2^a]) \leq \frac{2^{a+1} - 2r_{\min}}{2^{a-12}}$, - $\mathcal{P}([2^j, 2^{j+1}]) \leq \frac{2^{j+1} - 2^j}{2^{j-12}} = 2^{12}$, - $\mathcal{P}([2^{b-1}, r_{\max}]) \leq \frac{2r_{\max} - 2^b}{2^{b-12}}$.

Therefore,
$$\mathcal{P}(\mathcal{E}) \leq \frac{2^{a+1} - 2r_{\min}}{2^{a-12}} + 2^{12}(b - a - 1) + \frac{2r_{\max} - 2^b}{2^{b-12}}.$$

After simplification,
$$\mathcal{P}(\mathcal{E}) \leq 2^{12}(b - a) + 2^{13-b} r_{\max} - 2^{13-a} r_{\min}.$$

Thus, we obtain the upper bound

$$\boxed{\mathcal{P}([r_{\min}, r_{\max}]) \leq 2^{12}(b - a) + 2^{13-b} r_{\max} - 2^{13-a} r_{\min}}$$

### G.3. Accessibility of Sequences for Different Models

We follow the cramming protocol described in Section 5.1. Following (Kuratov et al., 2025), we optimize the soft prompts using the Adam optimizer for 3000 steps with a learning rate of 0.01, and a weight decay of 0.01. We run the optimization on a server with 2 NVIDIA H100 GPUs.

---

[2]Transformers are often trained using Brain Floating Point (bf16), but since it has fewer precision bits ($p = 8$), it implies a larger $\varepsilon$. We therefore use fp16 as a conservative choice.

Figure 7 reports results on the Pythia model suite at three scales with matched architecture (A: 160M, B: 410M, C: 1B). Across all model sizes, accessibility decreases sharply as sequence length increases, while the derived $n_{50}$ values grow approximately linearly with the number of memory vectors $m$.

Figure 8 compares different model architectures (A: Qwen-2.5, B: Gemma-3, C: Llama-3.2). While the accessibility curves for Gemma-3 and Llama-3.2 deviate from a clean exponential decay, all architectures still exhibit an approximately linear relationship between $n_{50}$ and $m$.

### G.4. Support of Different Models

We estimate support parameters for three geometries (Ball, Cone, and Ellipsoid) by Monte Carlo using the sampling procedure described in Section 5.2. For each sequence length $\ell \in 1, \ldots, \ell_{\max}$, we draw $N = 10^5$ i.i.d. token sequences $\mathbf{t}^{(k)} = [t_1, \ldots, t_\ell]$, where tokens are sampled independently and uniformly as $t_i \sim \mathcal{U}(\{1, \ldots, |\mathcal{V}|\})$. Each sequence is mapped to an embedding vector $Y^{(k)} = (L(\mathbf{X}^{(k)}))_{:,-1} \in \mathbb{R}^d$, where $\mathbf{X}^{(k)}$ is the embedded input of $\mathbf{t}^{(k)}$.

For the Ball baseline, we estimate the $\ell_\infty$ radius

$$r = \max_k \|Y_k\|_\infty.$$

For the Cone baseline, we additionally estimate the minimum pairwise cosine similarity to find the opening angle $\theta$

$$c_{\min} = \min_{i<j} \frac{\langle Y_i, Y_j \rangle}{\|Y_i\|_2 \|Y_j\|_2}, \qquad \theta = \arccos(c_{\min}).$$

All support parameters are estimated independently for each $\ell$. We verified empirically that increasing $N$ beyond $10^5$ does not materially change the estimates.

Given $(r, \theta)$, together with the embedding dimension $d$, vocabulary size $|\mathcal{V}|$, and precision $\varepsilon$ defined in Appendix G.2, we compute the theoretical slope proxies used in our plots:

$$C_{\text{ball}} = \frac{d \log(1 + 2R/\varepsilon)}{\log |\mathcal{V}|}, \qquad C_{\text{cone}} = \frac{d \log(1 + 2R/\varepsilon) + \log \text{FracCone}(\theta; d)}{\log |\mathcal{V}|}.$$

with $\text{FracCone}(\theta; d)$ defined as the ratio of packing numbers in Corollary F.5. For the Ellipsoid (axis-aligned rectangle) baseline, we estimate per-coordinate ranges $r_j = \max_{k,k'} |(Y_k)_j - (Y_{k'})_j|$ and use

$$C_{\text{elli}} = \frac{\sum_{j=1}^{d} \log(1 + r_j/\varepsilon)}{\log |\mathcal{V}|},$$

Figure 9 reports $C_{\text{ball}}$, $C_{\text{cone}}$, and $C_{\text{elli}}$ estimated from samples $\mathbf{X}_k$ with increasing maximum sequence length $\ell_{\max}$. For all models, the estimated constants stabilize rapidly as $\ell_{\max}$ increases, indicating that short sequences are sufficient to obtain stable estimates of the slope upper bounds.

## H. Computing the Cell Volume Distribution

In order to compute the cell volumes $|\mathcal{E}_t|$ corresponding to each token in Section 5.3, we use the ellipsoid support estimate from Section 5.2 and sample points uniformly in volume inside this ellipsoid. (i.e. we sample $\mathbf{Y}^{(k)} \sim \mathcal{U}([\mathbf{x}_1^{\min}, \mathbf{x}_1^{\max}] \times \cdots \times [\mathbf{x}_d^{\min}, \mathbf{x}_d^{\max}])$) For each sampled point $\mathbf{Y}^{(k)}$, we compute its greedy token $\hat{t}^{(k)} = \sigma(\mathbf{F}\mathbf{Y}^{(k)})$ via the decoder and count how often each token is selected. The relative volume of token $t$ is estimated by

$$|\mathcal{E}_t|/|\mathcal{E}| \approx \frac{1}{N} \sum_{k=1}^{N} \mathbb{I}\{\hat{t}^{(k)} = t\},$$

where $\mathcal{E}$ is the support ellipsoid and $N$ is the number of samples (here $N = 50\text{M}$).

Figure 10 shows the ranked cell-volume profiles. For most models, volumes are extremely uneven: a small set of tokens takes up a large fraction of the support, while the vast majority of tokens have tiny cells. Gemma-3 (and, to a lesser extent,

*Table 3.* Tokens with the largest estimated decoder-cell volumes (Section 5.3). Volumes are shown in parentheses. Whitespace/control characters are rendered explicitly (e.g., ␣, \n). For non-ASCII tokens (e.g., Gemma), we show the Unicode code.

| Model | Tokens corresponding to largest cells $\mathcal{E}_t$ ($|\mathcal{E}_t|/|\mathcal{E}|$) |
|---|---|
| Pythia-160M | `<ν>` (5.5e−3), `<κ>` (4.7e−3), `<␣>` (4.1e−3), `<\|endoftext\|>` (3.9e−3), `<well>` (2.9e−3) |
| Pythia-410M | `<\n>` (0.21), `<,>` (0.08), `< (` (0.07), `<->` (0.06), `<.>` (0.05) |
| Pythia-1B | `<\n>` (0.25), `<,>` (0.21), `<->` (0.10), `<.>` (0.054), `< (` (0.036) |
| Llama-3.2 1B | `<␣>` (0.38), `<,>` (0.19), `<\n>` (0.094), `< (` (0.059), `<->` (0.051) |
| Qwen-2.5 0.5B | `<␣>` (0.70), `<,>` (0.14), `<->` (0.038), `<\n>` (0.036), `<.>` (0.021) |
| Qwen-2.5 1.5B | `<␣>` (0.81), `<,>` (0.076), `<\n>` (0.038), `<->` (0.033), `< (` (0.011) |
| Gemma-3 270M | `<eos>` (1.1e−4), `<U+000A U+000A>` (1.0e−4), `</>` (9.6e−5), `<U+3054 U+7406 U+89E3>` (7.0e−5), `<<h2>>` (6.7e−5) |

Pythia-160M) looks noticeably more uniform. One possible explanation is their large vocabulary size compared to their number of parameters: if a model has many rare tokens, their decoder embeddings may be weakly trained and remain closer to random. This can lead to a more even partition of the embedding support, with fewer overwhelmingly large cells. As a sanity check, Table 3 shows that the largest-volume cells in most models align with very frequent tokens (whitespace, punctuation, and common function words). In contrast, Gemma-3 and Pythia-160M have top-volume tokens that appear less obviously frequent, which is consistent with the vocabulary explanation above.

We estimate the distribution of subsequent sequences by using the previously estimated next-token distribution $\mathcal{D}$, and then model the distribution over length-$n$ continuations using the $n$-fold multiplicative convolution $\mathcal{D}^n$. Figure 11 shows the resulting volume-cell distributions for sequence lengths $n \in \{1, \ldots, 6\}$ across models. Most models exhibit highly skewed next-token distributions, leading to a rapid concentration of mass near zero as $n$ increases. Models with more balanced distributions, such as Gemma3-270m, exhibit a slower decay.

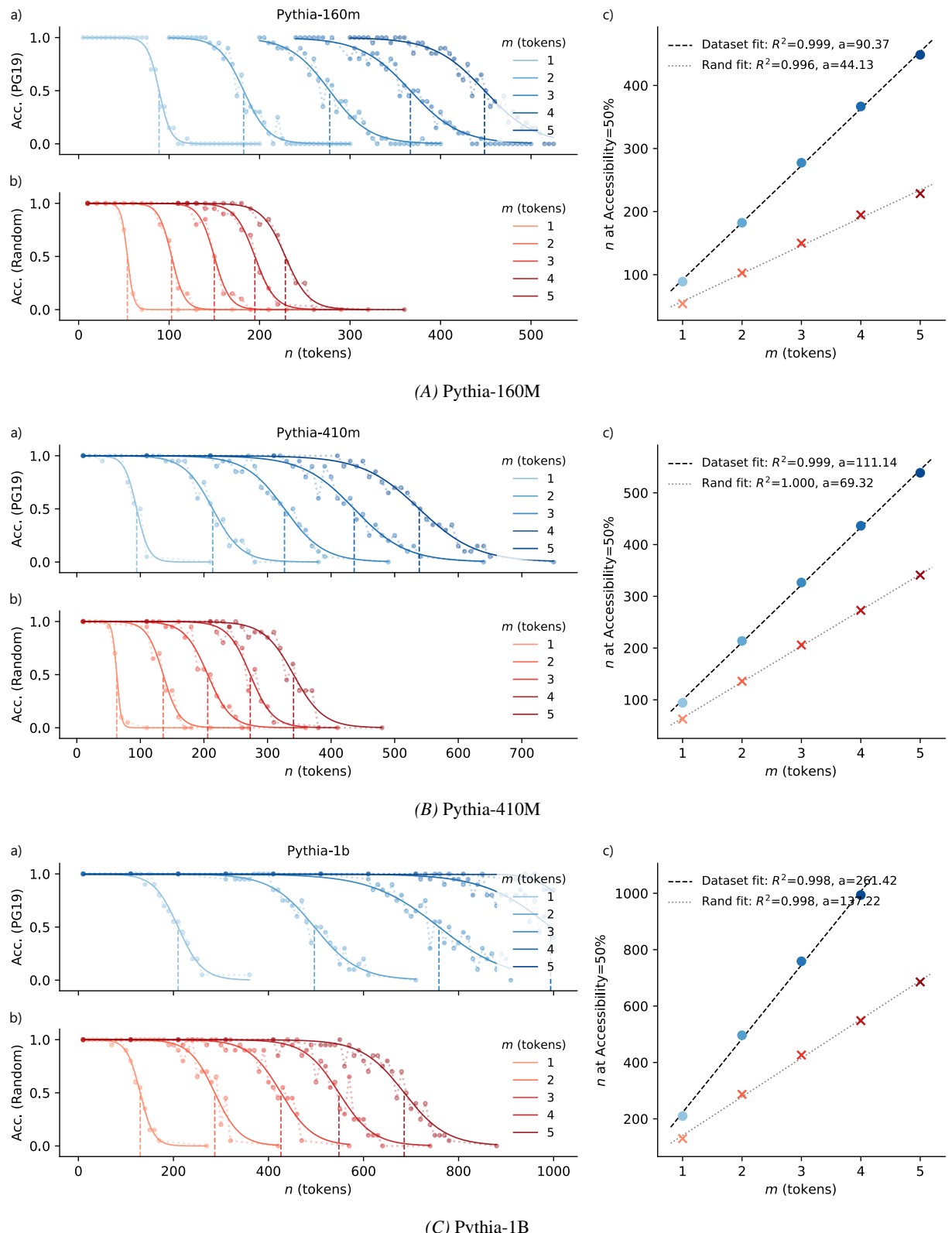

*Figure 7.* Mean accessibility for a) PG19 and b) random target sequences of length $n$ as a function of the number of trainable memory vectors $m$. For each $m$, we fit a sigmoid (solid) and mark $n_{50}$ where the fit crosses 0.5 (vertical dashed line). (c) $n_{50}(m)$ for PG19 (blue) and random (red), with linear fits (dashed). Results shown for the Pythia model suite at different scales A) 160M, B) 410M, C) 1B.

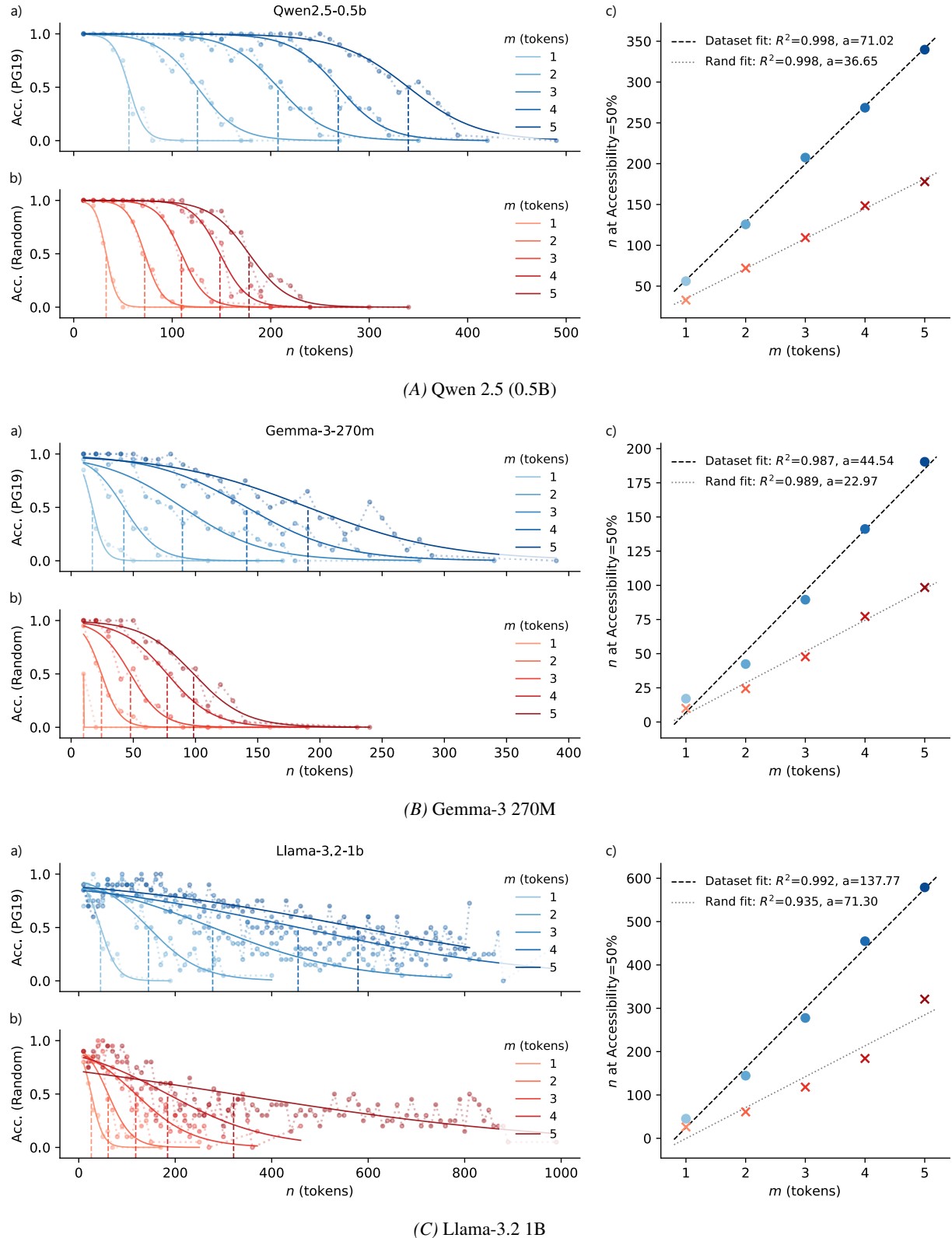

*Figure 8.* Mean accessibility for a) PG19 and b) random target sequences of length $n$ as a function of the number of trainable memory vectors $m$. For each $m$, we fit a sigmoid (solid) and mark $n_{50}$ where the fit crosses 0.5 (vertical dashed line). (c) $n_{50}(m)$ for PG19 (blue) and random (red), with linear fits (dashed). Results shown for three architectures A) Qwen-2.5, B) Gemma-3, C) Llama-3.2.

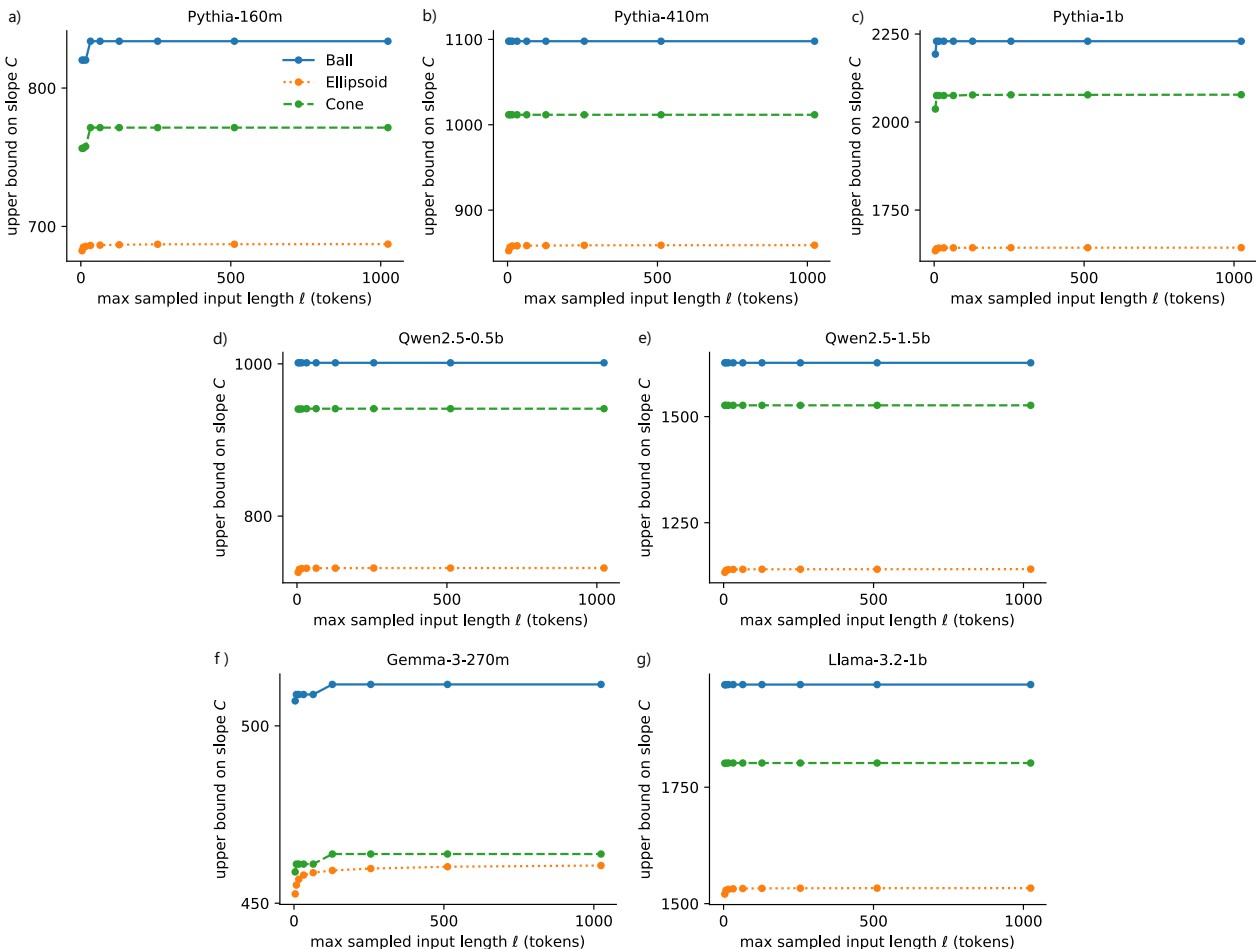

*Figure 9.* Upper bound on $C$ for different support geometry —Ball (blue), Cone (green), Ellipsoid (orange)—, estimated using 10K randomly sampled input strings of maximum length $\ell$. Sampling prompts longer than $\ell \approx 500$ suffices to estimate the upper bound. Pythia models for different sizes: a) 160M b) 410M, c) 1B and support for different model architectures d–e) Qwen (0.5B, 1.5B) f) Llama 1B, g) Gemma 270M.

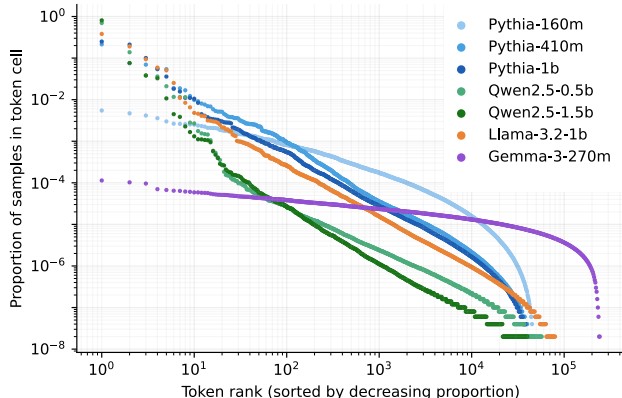

*Figure 10.* Relative volumes of decoder cells across tokens (log-log). For each model, tokens are sorted by estimated cell volume (largest on the left). A small set of tokens (typically $< 10^2$) accounts for a large fraction of the support volume, while most tokens ($10^4$–$10^5$) have tiny individual volumes (often $10^{-6}$–$10^{-8}$ of the support).

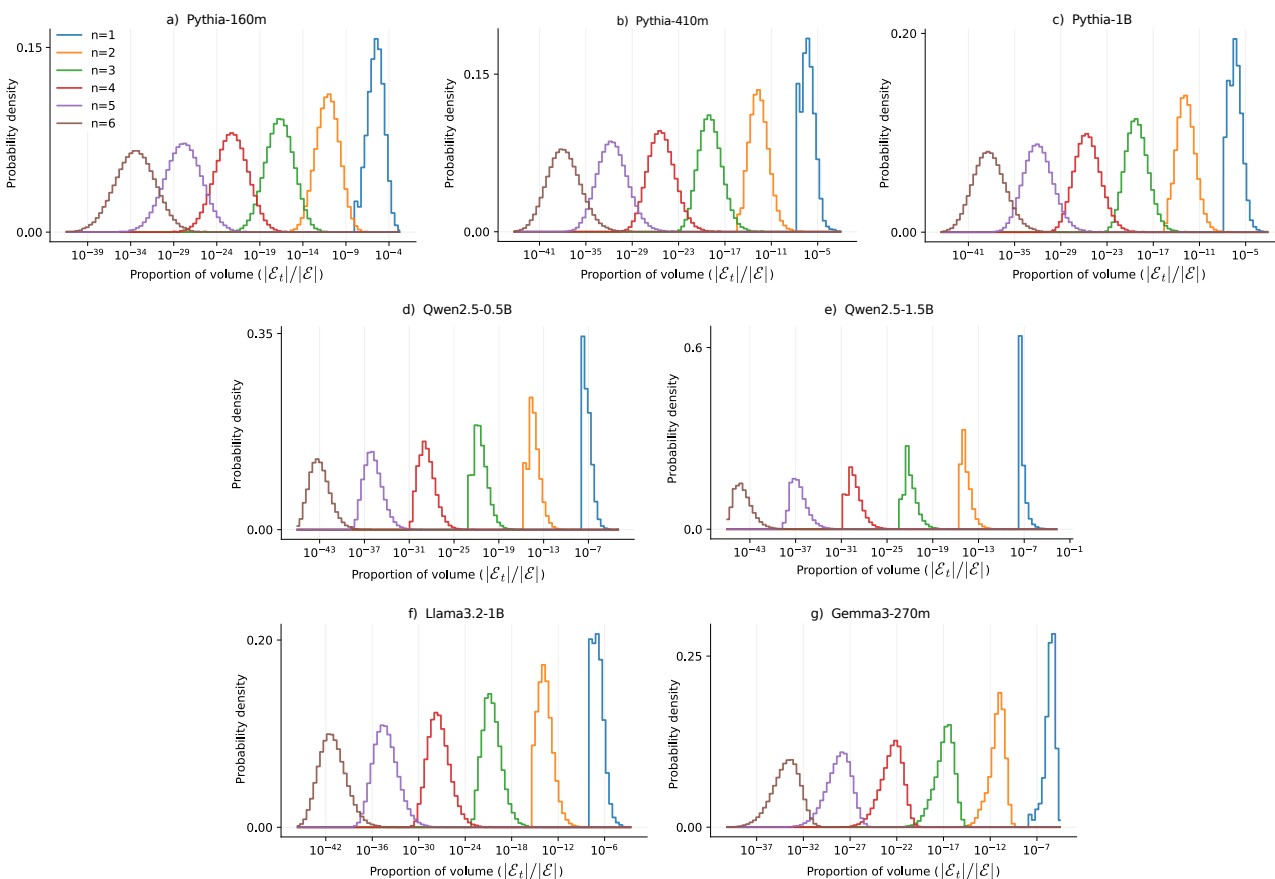

*Figure 11.* Distribution of $n$-sequences volumes proportions by estimating the $n$-fold convolution of the empirical one-step volume distribution $\mathcal{D}$ for different models: a) Pythia-160M, b) Pythia-410M, c) Pythia-1B, d) Qwen2.5 0.5B, e) Qwen2.5 1.5B, f) Llama3.2 1B, g) Gemma 270M.

