# OpenReview forum: "How Many Different Outputs Can a Transformer Generate?"
_ICML.cc/2026/Conference — ICML 2026 spotlight_

### Official Review · Reviewer_wrHG · 2026-03-11

**Soundness:** 4
**Presentation:** 3
**Significance:** 2
**Originality:** 2
**Overall Recommendation:** 4
**Confidence:** 4

**Summary:**

This paper investigates the intrinsic limitations of transformers in generating long, exact target sequences (such as exact copying tasks). While the authors frame this as a study of "expressivity," the core contribution is more accurately a rigorous formalization of sequence accessibility under the constraints of finite numerical precision and embedding space geometry. The authors argue that the well-documented empirical failures of transformers on simple long-sequence algorithmic tasks are not mere training artifacts, but hard structural limits. They show this first by proving that transformers with absolute positional encodings cannot output arbitrary sequences with length $>n$ when that exceeds a high value (dependent on the input length $m$). The authors then show that this bound can be made independent of the input prompt $m$ by using a mean field formulation (that looks at the transformers as architectures that map between probability measures). By modeling finite precision via packing numbers and modeling the unembedding matrix as a set of partitioning hyperplanes, the authors prove that:
- Finite Access: Every finite-precision transformer can only access a finite set of output sequences.
- Abrupt Failure Threshold: For a given soft prompt of length m, the ability to access a target sequence of length n drops exponentially fast once n exceeds a specific, mathematically derivable threshold.
- Absolute Limits: Using Wasserstein distance over the space of empirical measures, they establish an absolute upper bound on sequence accessibility that holds even if the prompt length is unbounded.
Finally, the authors empirically validate their theoretical bounds in experiments. Using a cramming setup (optimizing continuous soft prompts for frozen pretrained models like Qwen, Llama, and Pythia, as one in past work), they demonstrate that real-world models exhibit the exact exponential decay in accessibility predicted by their theorems. By refining their geometric assumptions to account for different embeddings and non-uniform token distributions, they show their theoretical threshold formula closely predicts empirical failure points.

**Compliance With Llm Reviewing Policy:**

Affirmed.

**Final Justification:**

I have increased my score to 4 from 3. My main concerns with the framing of the paper, and how it was positioned. While the work done by authors has merit, the way it was framed around expressivity was ambiguous to the point of being incorrect. However, with proper contextualization of the current draft, which the authors have promised to do, and apt discussion about the results wrt copying, I would be okay with the paper being accepted. I am still not giving it a full accept (5/6), as these critical changes (framing, important discussions) will not get reviewed, but I am leaning towards accept.

**Key Questions For Authors:**

- Impact of Positional Encoding Schemes (APE vs. RoPE): The theoretical framework in Section 2 (specifically Definition 2.2) explicitly models Absolute Positional Encodings (APE) as a simple additive matrix P at the input layer. However, the empirical validations in Section 5 utilize modern LLMs (such as Llama-3.2, Qwen-2.5, and Gemma) which rely heavily on Rotary Positional Embeddings (RoPE). Because RoPE applies rotational, relative transformations at every attention layer rather than a single additive offset at the embedding layer, how does this impact the geometric volume bounds? Does the packing number argument gracefully and naturally extend to RoPE without modification?

- The Assumption of Weight Tying: In Definitions 2.5 and 2.6, the theoretical transformer assumes weight tying, explicitly using the transposed input embedding matrix $E^\top$ as the final linear projection to calculate logits. If a model utilizes untied weights (a completely separate unembedding matrix), does this alter the argmax partitioning of the embedding space described in Section 3.1? How would the theoretical upper bounds shift if the output hyperplanes are no longer strictly constrained by the input embedding geometry?

- The "Goodness" of Inaccessibility: The paper largely frames the exponential decay of sequence accessibility as a negative limitation (a "failure" to copy or generate). However, drawing a parallel to statistical learning theory and concepts like VC dimension, an architecture capable of accessing and memorizing any arbitrarily long target sequence would possess functionally infinite capacity, which often leads to severe overfitting and a lack of generalization. Do the authors agree that this mathematically bounded accessibility might actually serve as a necessary regularizer or structural inductive bias for transformers? I would appreciate a brief discussion in the text contextualizing whether this "failure" is actually a necessary feature (and thus a good thing) of finite-precision learning.

**Limitations:**

So, my main concerns about the paper are that, while the results of the paper are important, the way they are framed are slightly misleading. Right from the title to the introduction to the final conclusion/takeaway, there is scope to reframe and state the contributions of the paper more accurately.

**Strengths And Weaknesses:**

**Strengths**

*Theoretical Framework*: The paper uses mean-field theory (representing prompts as continuous probability distributions) with high-dimensional geometry and packing numbers. This allows the authors to formulate mathematically rigorous, prompt-independent bounds on transformer generation, which is a relevant theoretical result.

*Strong Empirical Validation*: The experiments in Section 5 confirm specific the theoretical predictions and give credence to the theory stated in the paper.
​
*Methodological Rigor in Bounds*: The authors do not stop at their initial, loose theoretical upper bound. By refining their assumptions to account for the real-world transformers and testing their theory on LLMs, the authors have done a good job in trying to show how their theory would matter in practice.

**Weaknesses**

*Misaligned Terminology and Missing Literature ("Expressivity")*: The paper frames itself as a study of transformer "expressivity." However, in the theoretical study of transformers, expressivity is a heavily loaded term and typically referred to formal language theory and the in-principle ability of the architecture to do a task (irrespective of whether it can be learnt). There is a massive body of literature on this exact topic—often explicitly considering finite precision—that is entirely absent from the related works. In fact, the way the authors use expressivity is also different from this definition I just mentioned.
- Feedback: The authors must engage with and differentiate their work from foundational papers on transformer expressivity and formal languages. I will just cite a survey - Strobl et al' [1] as that might be the best place to find most of these papers. But for concrete examples -- Hahn & Rofin [2,6,7], Yang & Chiang's work on BRASP/CRASP [3,4,5] and Will Merrill's work (the authors cite one paper, but avoid a detailed contextualization of their paper's finding). I am providing that many citations to just communicate the extent of how big of a gap it is to talk about expressivity and not talk about such papers. To avoid confusion, the paper should reframe its core claim around "generative accessibility" or "capacity" rather than expressivity.

*Disconnect Between the Motivation (Copying) and the Proofs*: The paper motivates its analysis by citing the failure of transformers on copying tasks (Jelassi et al [10]). However, the theoretical results treat all sequences of length $n$ as essentially equal victims. The proofs show that because the number of possible sequences vastly outgrows the packing number of the prompt space, most sequences become inaccessible. However, the theorem proves that some (and eventually most) sequences are inaccessible, but it does not prove that the specific sequence a user wants to copy is inaccessible. If a prompt can only access 1 sequence, but it is the correct sequence, the task succeeds.
Furthermore, past work has shown ([7,8,9]), that unique token copying length-generalizes quite well, whereas repeated copying fails abruptly. In fact [7] gave theoretical reasons why this might be the case. The current volumetric/packing argument is too blunt to explain this structural discrepancy. The authors need to better contextualize how the internal structure or compressibility of the target sequence interacts with their capacity bounds, because failure on copying is not explained here properly.

[1]Hahn, M. (2020). Theoretical limitations of self-attention in neural sequence models. Transactions of the Association for Computational Linguistics

[2] Strobl, Lena, et al. "What formal languages can transformers express? a survey." Transactions of the Association for Computational Linguistics 12 (2024): 543-561.

[3]Chiang, David, Peter Cholak, and Anand Pillay. "Tighter bounds on the expressivity of transformer encoders." International Conference on Machine Learning. PMLR, 2023.

[4] Yang, Andy, and David Chiang. "Counting like transformers: Compiling temporal counting logic into softmax transformers."

[5] Yang, Andy, David Chiang, and Dana Angluin. "Masked hard-attention transformers recognize exactly the star-free languages." Advances in Neural Information Processing Systems 37 (2024): 10202-10235.

[6] Hahn, Michael, and Mark Rofin. "Why are sensitive functions hard for transformers?." Proceedings of the 62nd Annual Meeting of the Association for Computational Linguistics (Volume 1: Long Papers). 2024.

[7] Huang, Xinting, et al. "A formal framework for understanding length generalization in transformers. ICLR'2025

[8] Zhou, Hattie, et al. "What algorithms can transformers learn? a study in length generalization. 2023

[9] Jobanputra, Mayank, et al. "Born a Transformer--Always a Transformer? On the Effect of Pretraining on Architectural Abilities. NeurIPS'2025

[10] Jelassi et al. "Repeat after me: Transformers are better than state space models at copying". ICML'2024

---

> ### Author Rebuttal · Authors · 2026-03-29
>
> **Misaligned Terminology and Missing Literature.**
>
> We thank the reviewer for pointing this out and for the helpful references.
>
> We propose revising the title to ``How Many Different Outputs Can a Transformer Generate?'' to avoid confusion, and have revised the framing throughout the paper to emphasize accessibility and capacity rather than expressivity.
>
> We have also added a discussion comparing our results to prior work on transformer expressivity from the perspective of formal language theory. The key distinctions are:
>
> - *Output Structure*: In formal language settings, outputs are typically binary as the goal is to determine membership in a language. By contrast, our analysis considers sequence outputs. For tasks such as copying, distinguishing between inputs is essential, as different inputs necessarily lead to different outputs. In contrast, in formal languages, a model does not need to distinguish between two inputs if both are (respectively aren't) in the language.
>
> - *Quantitative vs. qualitative*: Our main contribution is to provide quantitative bounds on the proportion of accessible sequences, which we show to be tight through cramming experiments. This contrasts with formal language results, which are typically qualitative (i.e., whether a word belongs to a language).
>
>  - *Practical Implications*: The first two points lead to practical impact complementary to the conclusions drawn from transformer expressivity work. One of them is to theoretically characterize the threshold effect in the crammimg task. Another regards test-time training, and we refer to our response to Reviewer 24hJ in that regards.
>
>  - *Generality*: As noted by Reviewer 24hJ, our results extend beyond transformers to any architecture with bounded representations and finite precision, whereas results in formal language theory are typically tied to specific computational models.
>
>  - *Precision scaling*: We discuss this point in our answer to the last question of Reviewer 24hJ.
>
> **The theorem [...] does not prove that the specific sequence a user wants to copy is inaccessible.**
>
> This is correct. Our results predict a drop in *average* accuracy on the copying task, rather than failure on every instance.
>
> Importantly, the set of accessible sequences is independent of the prompt. As a result, only a fixed subset of sequences can be generated, regardless of the input. Since the number of possible sequences grows exponentially with length, the probability that a randomly chosen target sequence lies in this accessible set quickly vanishes.
>
> This remains true even when restricting to natural language sentences, whose number still grows exponentially with length.
>
> **Past work has shown ([7,8,9]), that unique token copying length-generalizes quite well, whereas repeated copying fails abruptly. [...] The authors need to better contextualize how the internal structure or compressibility of the target sequence interacts with their capacity bounds.**
>
> First, we note that the setting of unique copying is inherently bounded: since each token can appear at most once, the maximum sequence length is limited by the vocabulary size. While one may still observe a form of length generalization within this range, it is fundamentally different from the regime considered in our work.
>
> In addition, a finer characterization of structure is challenging. This is evidenced by the work cited by the reviewer [7,8,9], which only differentiates between the fully general copying task, and the extremely restrictive unique copying one.
>
> That said, our framework does incorporate structural effects. In particular, Figure 3 illustrates the distribution of sequence ``hardness,'' measured by the volume each sequence occupies in representation space. By leveraging this distribution (rather than assuming uniform cell volumes), we obtain tighter bounds that reflect variability across sequences.
>
> **Impact of Positional Encoding Schemes.**
>
> In our framework, $\mathbf X$ denotes the input after positional encoding. As a result, our analysis is agnostic to the specific positional encoding scheme, and the packing-based arguments apply without modification.
>
> The choice of positional encoding may still have an indirect effect on the bounds through the geometry of the representation space.
>
> **The Assumption of Weight Tying:**
>
> As in the case of positional encoding, we consider $\mathbf X$ to be the representation after embedding. As a result, using untied output weights does not affect the validity of our analysis.
>
> **The ``Goodness'' of Inaccessibility:**
>
> We agree with the reviewer’s intuition that this can be a desirable property, and we believe it is largely unavoidable in finite-precision learning.
>
> A key aspect of our analysis is that it does not rely on specific architectural details of transformers, but only on two general properties: finite precision and bounded representations. This suggests that such ``failures'' are in fact a necessary feature of these systems.

---

> > ### Author Rebuttal · Reviewer_wrHG · 2026-03-31
> >
> > Like I mentioned in my comment, my main concerns were about framing, some missing interaction with existing literature and treatment of copying. Overall with the promises made by the authors to rename the paper and to add apt discussion sections, I am happy with their response.

---

### Official Review · Reviewer_hRzw · 2026-03-11

**Soundness:** 3
**Presentation:** 2
**Significance:** 3
**Originality:** 3
**Overall Recommendation:** 5
**Confidence:** 4

**Summary:**

The paper considers a particular type of expressivity limitations of transformers: given a prompt of length m, what is the maximum length n of an accessible sequence, i.e., a sequence the model can generate? There are two core theoretical results:
1. following from finite precision and the assumption that embeddings are within a sphere of a fixed radius, the authors show accessible sequence length n is linear in prompt length m.
2. with an additional, stronger assumption about Wasserstein precision, the authors show an upper bound on accessible sequence length n independent of m, i.e., showing that adding more prompt tokens to the kv cache will not allow the model to generate longer sequences.
Additionally, the paper empirically tests the relationship between m, n using cramming methodology (optimizing a soft prompt to reconstruct a sequence). The findings suggest a linear relationship, in line with result 1.

**Compliance With Llm Reviewing Policy:**

Affirmed.

**Key Questions For Authors:**

## Questions

The main result that the number of sequences that can be generated scales linearly with prompt length does not actually predict failures on copying, since there the output scales linearly with the prompt (sequence that should be copied). On the other hand, the previously mentioned line of work on length generalization in terms of C-RASP definability predicts these copying issues. Can you comment more on the implications of your results for copying and related tasks?

> In addition, the theoretical slope from Theorem 4.5 is a close upper bound on the slopes estimated from Fig. 2c (first row of Table 1).

It's currently hard to tell the differences. Mention in text that the factor is between 5-10

**Limitations:**

One thing that is unclear to me is how reasonable assumption 4.7 about Wasserstein precision is (see Weaknesses for more info).

**Strengths And Weaknesses:**

## Strengths

This paper has a clearly scoped contribution with interesting theoretical and empirical results. The analysis of expressivity in terms of accessible sequences complements existing analysis of transformer expressivity in terms of formal languages, where the set of prompts that can be recognized by a transformer is typically the object of study.
## Weaknesses

### Wasserstein Precision Assumption

Wasserstein precision should be more clearly introduced. In Assumption 4.7, it is currently left generic as Wq without a definition, informal description, or reference.

Moreover, how reasonable do you take the Wasserstein distance assumption to be? Empirically, you find a linear relationship between m and n, in line with result 1. I'm left wondering whether you think this trend would break down for large enough m (i.e., you enter the regime of result 2), or whether result 2 may rely on assumptions that are too strong.

### Engagement with Existing Work on Transformer Expressivity

Section 1.1 misses or only loosely characterizing the large body of existing work on transformer expressivity in terms of formal languages (see Strobl et al for a survey).

In particular, this line of work shows limitations on the functions computable by transformers in terms of complexity classes. In particular, there's a divide between fixed-precision transformers, close to AC0 (cf. https://arxiv.org/abs/2503.14615 inter alia), and log-precision transformers, corresponding to TC0 (cf. https://arxiv.org/abs/2409.13629 inter alia). Beyond acknowledging the fact work, it would be interesting to discuss your results in the context of these two different regimes of transformers, since your work is also dealing with a transition based on precision in a sense. Your main results clearly depend on the precision. In the high-level discussion of the results, it would be natural to expose the role of precision. For instance, in relation to the previously mentioned results, if you allow precision to scale O(1) vs. O(log m), how does this change the accessible sequence length n?

Another existing line of work that would be worth engaging with is the C-RASP Conjecture (Yang et al): a hypothesized explanation for the types of tasks where transformers can length-generalize, which seems extremely relevant to this work and important to engage with. Are there connections or different predictions between the two?

Finally, the current title is very general as expressiveness of transformers is essentially an entire research area. Many expressivity limitations unrelated to this work have already been identified, eg using circuit complexity, as mentioned above. If it is possible to revise the paper title at this stage, I would advise choosing something more precise involving accessible sequences.
### Presentation and Clarity Issues

In general, the proofs of the main results are not clearly presented, even though the high-level idea, at least in result 1, is quite simple: finite precision in a bounded region will bound the number of sequences that can be distinguished. The paper would be significantly improved by cleaning up the proofs to highlight the concepts at work. The part of the paper that introduces different concepts is also sometimes unclear and inconsistent, which I discuss in more detail below.

> corresponds to the set of last embedding tokens that most likely leads to the next token ti.

I believe you mean Ei is the set of last embedding vectors that most likely lead to next token ti

Section 3.2: notation is unclear and possibly incorrect

In describing your multi-token method, you should clarify what a prompt means for you. It appears X is meant to be a soft prompt of length m, but this simple detail is not immediately clear.

Related to this, what's the connection between t and y? It seems y is the embedding of the sequence t?

Between sections, you seem to have flipped notations. In section 3, X is the prompt, but in section 4, you say that a sequence X is accessible if there exists some prompt that makes X likely to generate. Then definition 4.1 says X is the prompt again. Please fix these inconsistencies!

"all sentences of length n >= N" => sequences or strings

In Section 3, clarify your statement about embedding radius. Is r meant to be fixed as you increase embedding width or depth? If it's meant to hold across widths, the parameterization will be relevant for whether this holds at initialization and beyond (see muP and completeP literature). Normalization is also relevant for justifiying this boundedness assumption: with appropriate initialization, you could likely show this analytically based on the presence of layer-norm and the fact that other operations are Lipschitz.
### Slight Correction to Claim About Sequence Likelihood

> the set of soft prompts that most likely leads to the sequence t

This is imprecise and not exactly correct. What your definition specifies is the set of prompts that lead to t under greedy decoding. This is not the same thing as the set of prompts where the most probability is placed on t via a measure of strings. Consider the case where generating t1 has low but nonzero probability, and t2, ..., n are fully deterministic. Then greedy decoding would not generate this string, but for large n, it would be the "most likely" string generated from the model in the sense that the probability measure over strings induced by the model assigns it the highest probability.

---

> ### Author Rebuttal · Authors · 2026-03-29
>
> We thank the reviewer for their constructive feedback, which we have carefully addressed below.
>
> **Wasserstein precision should be more clearly introduced.**
>
> While both Wasserstein distance and precision are defined in the manuscript, their definitions were too far apart (lines 108 and 296).
>
> We have revised the manuscript to recall the definition of $W_q$ at Assumption 4.7 and to clarify the definition of Wasserstein precision.
>
> **Moreover, how reasonable do you take the Wasserstein distance assumption to be?**
>
> We agree that the Wasserstein distance assumption can be debated. For this reason, we introduce an alternative assumption in Appendix D, which we believe is more natural.
>
> In particular, it aligns more closely with how semantic similarity is perceived in language: modifying a small fraction of a large text’s tokens, while preserving both the beginning and the end, typically does not change the meaning of the sequence.
>
> **On the linear relationship between m and n.**
>
> That is a very good point. Since cramming experiments become very expensive for long sequences, we weren't able to verify the break down of the linear scaling through this task.
>
> To address this concern, we added a new copying-length generalization experiment following [1]. We fine-tuned pretrained Pythia-1B and Pythia-1.4B models on a synthetic exact string-copying task using next-token cross-entropy loss. Each training example was constructed from a random string $x$ of length at most 50, formatted as $x|x$, where the model was given the prefix $x|$ and trained to generate the target copy $x$. Training was stopped once exact-match accuracy at length 50 reached 100\%, or after 10K training steps.
> We then evaluated exact-match copying accuracy on longer unseen strings within the model context window.
> Results are
> | Model/Eval Length | 50 | 100 | 200 | 300 | 400 |
> | -- | ---: | ---: | ---: | ---: | ---: |
> | Pythia-1.4B | 1.00 (0.00) | 0.92 (0.06) | 0.04 (0.06) | 0.00 (0.00) | 0.00 (0.00) |
> | Pythia-1B   | 1.00 (0.00) | 1.00 (0.00) | 0.42 (0.06) | 0.08 (0.06) | 0.00 (0.00) |
>
> We observed that performance remained near 100\% for shorter strings, then dropped sharply beyond a threshold, captured well by sigmoid fits (Pythia-1B: $R^2 = 0.97$, Pythia-1.4B: $R^2 = 0.96$).
> This suggests that the linear regime does not continue indefinitely, which is consistent with Result 2 (if a sentence is not accessible to a transformer $\tau$, in particular it is impossible for $\tau$ to copy it).
> ## Engagement with Existing Work on Transformer Expressivity
> We thank the reviewer for pointing this out. We propose revising the title to ``How Many Different Outputs Can a Transformer Generate?'' to avoid confusion.
>
> We have also added a paragraph comparing our results to prior work on transformer expressivity, incorporating the references suggested by the reviewer.
>
> **On the role of precision.**
>
> We refer to our response to the last question of Reviewer 24hJ, where we discuss the role of precision in detail.
>
> **On the C-RASP Conjecture.**
>
> While this line of work is indeed related, there are important differences, in particular regarding the role of output structure (binary vs. sequence outputs) and the quantitative nature of our bounds. We discuss these distinctions in more detail in our response to Reviewer wrHG.
> ## Presentation and Clarity Issues
> We thank the reviewer for these precise remarks and careful reading of the paper. We have made the suggested changes and answer the remaining questions below.
>
> **It seems y is the embedding of the sequence t?**
>
> That is correct. We have revised Section 3.2 to make this connection explicit.
>
> **Clarify your statement about embedding radius.**
>
> We refer to our response to the first question of Reviewer 24hJ, where we clarify the definition of the embedding radius. In particular, it is defined uniformly across widths, and we now provide both a formal proof and empirical evidence for its existence.
> ## Claim About Sequence Likelihood
> We thank the reviewer for pointing this out, and have updated the wording accordingly.
> ## Questions
> **The main result [...] does not actually predict failures on copying, since there the output scales linearly with the prompt.**
>
> We agree that Theorem 4.5 does not predict failure on copying. Instead, this behavior is captured by Theorem 4.8 (based on the Wasserstein assumption) and Theorem D.3 (based on the elementary operations assumption), both of which establish limits that are independent of the prompt length.
>
> **Can you comment more on the implications of your results for copying and related tasks?**
>
> Our results provide a quantitative analysis of accessibility that applies across tasks such as copying, cramming, and test-time training. We refer to the Quantitative vs. qualitative and Practical Implications paragraphs in our response to Reviewer wrHG for more details.
>
> [1] Jelassi et al. ``Repeat after me: Transformers are better than state space models at copying''. ICML'2024

---

> > ### Author Rebuttal · Reviewer_hRzw · 2026-03-31
> >
> > The title change (How Many Different Outputs Can a Transformer Generate?) and proposed edits address my concerns around clarity, correctness, and engagement with past work. The new copying experiment is an interesting addition regarding the Wasserstein assumption. Thanks to the authors for thoughtfully engaging. I have raised my score and would like to see this paper accepted!

---

### Official Review · Reviewer_24hJ · 2026-03-13

**Soundness:** 4
**Presentation:** 4
**Significance:** 3
**Originality:** 4
**Overall Recommendation:** 5
**Confidence:** 4

**Summary:**

**Summary**
The paper studies the capacity of transformers to generate a desired target sequence given the choice of a prompt drawn from all prompts of a fixed length. There always exist target sequences which no prompt can generate. Furthermore, the proportion of generatable target sequences vanishes at an exponential rate when the target length is sufficiently large. The theoretical framework gives quantitative predictions for this rate of vanishing, which are shown to be remarkably accurate via experiments with transformer language models on a so-called “cramming” task.

**Compliance With Llm Reviewing Policy:**

Affirmed.

**Final Justification:**

The authors clarified my understanding that the theoretical results do not apply specifically to transformers, but suggested that it was not so general as to be a property of any possible model in practice. This motivated the specific focus on transformers.

I also second the suggestion to engage more critically with the expressivity literature -- however I do not think this is fatal, as the work is ultimately distinct.

Given the references to test-time training, promises to situate their work within the literature, and clarification of the assumptions, I will raise the rating of presentation and significance.

I do not give the paper a 6 because the theory currently still remains somewhat general/high-level and may not have so much explanatory power that explains phenomena beyond cramming.

**Key Questions For Authors:**

I would increase my rating of the paper’s presentation if (1) were addressed. I would increase my rating of the significance if (2) were addressed.

**Questions**
1. Is there any point in the analysis that depends on the transformer architecture? It seems like the bounds could be derived for any model that has finite precision and output contained inside a ball. I’m curious why this wasn’t generalized beyond transformers in the presentation.
2. Does this framework really explain failure on Copy? Given a fixed prompt length, the volume of copying sequences already occupies an exponentially vanishing subset of all sequences of length n. In particular, does this framework explain performance on any other task of interest besides cramming?
3. Do you see a way to make the same argument if the precision of the transformer was not fixed? For instance, if it was allowed to grow at a rate of log(n)?

**Limitations:**

yes

**Strengths And Weaknesses:**

**Strengths**
- A well-motivated question with an elegant answer
- Presented clearly. The intuition, as explained in figure 1 and the paragraph at line 208 was particularly insightful.
- The ability to make quantitative predictions on accuracy on the cramming task is striking.
- Approximating the support with ellipsoids and cones is an insightful extension of the results. There is room to improve the bounds in future work.

**Weaknesses**
- The results rely crucially on the assumptions, such as finite precision and the output being contained in a ball.
- The proof idea seems to apply to any model. It’s unclear whether the transformer architecture imposes any additional constraints.
- The framework explains performance on the cramming task, but applications to more fine-grained tasks seem uncertain.

---

> ### Author Rebuttal · Authors · 2026-03-28
>
> We thank the reviewer for their careful reading and clear understanding of our contributions.
>
> **The results rely crucially on the assumptions, such as finite precision and the output being contained in a ball.**
>
> These assumptions indeed play an important role in our analysis. At the same time, they are standard and naturally satisfied in practical transformer implementations:
>
> - Finite precision follows directly from the limited number of bits used in computation.
>
> - We assume that the intermediate representations at every layer lie within a ball of radius $r$. We validate it empirically in https://anonymous.4open.science/r/ICML-radius-bound/fig.png, showing via Monte Carlo sampling (as described in our paper) that representations lie in a fixed-radius ball independent of prompt length.
>
> As pointed out by Reviewer hRzw, we can also prove formally the latter assumption, which follows from the normalization operations (typically RMSNorm or LayerNorm) used in transformers, together with the continuous nature of the MLP and attention layers.
> Formally, the representation at layer $l+1$ is typically given by $$X^{l+\frac12}=X^l+\mathrm{Attention}(\mathrm{Norm}( X^l)),\quad
>  X^{l+1}=X^{l+\frac12}+\mathrm{MLP}(\mathrm{Norm}( X^{l+\frac12})),
> $$ where $\mathrm{Norm}$ is a normalization operator such that $\lVert\mathrm{Norm}(X)\rVert_\infty\leq N$. For a $l$-layer transformer, a natural upper bound is therefore $\lVert X^l\rVert_\infty\leq(I+k(A+M))\lVert E_f\rVert$, where the maximal initial norm $I=\sup\lVert X^0\rVert_\infty$ is entirely determined by the initial embedding matrix and positional encoding, $A=\sup_{\lVert X\rVert_\infty\leq N}{\lVert\mathrm{Attention}(X)\rVert_\infty}$, $M=\sup_{\lVert X\rVert_\infty\leq N}{\lVert\mathrm{MLP}(X)\rVert_\infty}$, and $E_f$ is the final embedding matrix (with the convention $\lVert X\rVert_\infty=\max_{i}\lVert X_i\rVert$).
>
>
> **Is there any point in the analysis that depends on the transformer architecture? It seems like the bounds could be derived for any model that has finite precision and output contained inside a ball.**
>
> This is entirely correct. Our bounds extend beyond transformers and apply to any architecture with bounded representations and finite precision.
>
> However, in state-space models such as Mamba, hidden states may diverge [2]. In such cases, one could argue that additional context information is stored in the growing norm of the hidden representation. While we believe our framework could still be adapted to this setting, doing so would introduce additional technical complexity and reduce clarity.
>
> For these reasons, we chose to focus on transformers, which remain the dominant architecture in practice.
>
> **Does this framework really explain failure on Copy? Given a fixed prompt length, the volume of copying sequences already occupies an exponentially vanishing subset of all sequences of length n.**
>
> We agree that, when restricting to meaningful natural language sequences, the set of valid outputs forms only a small subset of all possible sequences. However, this subset still grows exponentially with sequence length. As a result, our bounds in Theorems 4.5 and 4.8 continue to apply when the copying space is restricted to such sequences. This effectively amounts to replacing the denominator $|\mathcal V|^n$ with a smaller exponential term.
>
> **In particular, does this framework explain performance on any other task of interest besides cramming?**
>
> Another closely related setting is test-time training, as described in Section 3.1 (``Memory as Context'') of [1]. The goal of this approach is to mitigate the quadratic cost of attention with respect to context length by compressing a long context into a short sequence of vectors $h$, using a neural network $\mathcal M$. In this setting, $\mathcal M$ is trained to iteratively encode the context into $h$ via gradient-based updates. Our results provide direct insight into this process: they characterize how the size of $h$ must scale with the length of the context in order to preserve accessibility.
>
> **Do you see a way to make the same argument if the precision of the transformer was not fixed? For instance, if it was allowed to grow at a rate of log(n)?**
>
> We thank the reviewer for this insightful question.
>
> Allowing the precision to grow with $n$ (e.g., at rate $\log(n)$) would indeed increase the proportion of accessible sequences. However, Theorem 4.8 shows that sequences of length $n$ become inaccessible at precision $\varepsilon$ once $n > \frac{C}{\varepsilon^d}$ for some constant $C>0$. Therefore, ensuring accessibility for all sequences of length $n$ requires at least $\varepsilon \lesssim n^{-1/d}$, and a logarithmic increase in precision is insufficient.
>
>
> [1] Behrouz, A., et al. Titans: Learning to Memorize at Test Time. NeurIPS'25
>
> [2] Lu, P., et al. Mamba Modulation: On the Length Generalization of Mamba Models. NeurIPS'25

---

> > ### Author Rebuttal · Reviewer_24hJ · 2026-04-03
> >
> > Thank you for the insightful response!
> >
> > I see the point about copying. My specific question was more as follows: consider the copying task as taking prompts of length m and outputting a sequence which is the copy of the prompt. Theorem 4.5 says the proportion of accessible sequences decays exponentially as n grows, but this observation alone does not seem to suggest that copying is difficult (as the proportion of valid copy sequences already decays exponentially as n grows). I was curious about your response to this question, as the connection to copying is much less developed than the connection to cramming.
> >
> > The other reviewers mentioned making the title less general, and engaging with the previous literature on "expressivity" in terms of circuit classes and logic. I think this is a great idea. Your work and methods is distinct from this line of work, and that should be made clear.
> >
> > I think the paper presentation could also be improved by making reference to where these results apply (not just to transformers but all models having these general properties) and mentioning the fact about Mamba which shows that this is not so general as to be applicable to anything.
> >
> > Thanks again for an insightful paper and response. I will increase my rating of the significance and presentation (assuming the suggestions on presentation made by me and the other reviewers make it into the final draft). I maintain the score.

---

> > > ### Author Response · Authors · 2026-04-04
> > >
> > > We thank the reviewer for the insightful discussion, which greatly helped to improve the clarity and positioning of our work. We clarify the implications of our respective theorems below.
> > >
> > > Theorem 4.5 characterizes the linear scaling between prompt length $m$ and accessible output length $n$, and is therefore mainly relevant to the cramming setting.
> > >
> > > The limitation for copying instead follows directly from Theorem 4.8 (based on the Wasserstein assumption) and Theorem D.3 (based on the elementary operations assumption). The key point is that these results establish a global upper bound $N$ on the number of distinct sequences that a transformer can generate, independently of the prompt length. As a result, when the number of possible output sequences of length $n$ exceeds $N$, some sequences must be inaccessible. Moreover, the proportion of inaccessible sequences increases exponentially with $n$.
> > >
> > > In the paper, we apply this argument to the set of all sequences of length $n$ (of size $|\mathcal V|^n$). In Theorem 4.8, this leads to a threshold $$N_{\mathrm{all}}=\big(1+\frac{4r}{\varepsilon}\big)^d\frac{\operatorname{ln}\big(e + \frac{e(2r)^q}{\varepsilon^q}\big)}{\operatorname{ln}(|\mathcal V|)}.$$ The same reasoning can be applied to any subset of valid sequences, yielding a corresponding (greater) threshold $N_{\mathrm{valid}}$. For instance, if we take the set of valid sequences of length $n$ to be of size $10^n$, this yields $$N_{\mathrm{valid}}=\big(1+\frac{4r}{\varepsilon}\big)^d\frac{\operatorname{ln}\big(e + \frac{e(2r)^q}{\varepsilon^q}\big)}{\operatorname{ln}(10)}.$$ So our argument still holds as long as the amount of target functions of length $n$ grows exponentially with $n$. Then the decrease of the proportion of accessible sequences is still exponential, despite happening at a higher threshhold (e.g., $N_{\mathrm{valid}}$ instead of $N_{\mathrm{all}}$) and being less steep (e.g., $10^{-n}$ instead of $|\mathcal V|^{-n}$).
> > >
> > >
> > >
> > > Thanks again for the valuable comments, which we have all taken into account for the final version!

---

### Official Review · Reviewer_SYWu · 2026-03-13

**Soundness:** 3
**Presentation:** 2
**Significance:** 3
**Originality:** 3
**Overall Recommendation:** 4
**Confidence:** 2

**Summary:**

The paper investigates the fundamental expressivity limits of transformer models by formalizing the concept of accessible sequences—outputs that a model can potentially generate given some input prompt. The study establishes three primary theoretical results: (a) the maximal length of accessible sequences grows linearly with the prompt length; (b) beyond a critical threshold, the proportion of accessible sequences decays exponentially with sequence length; (c) these limits hold even with arbitrary prompt length. The empirical results confirm the linear scaling of accessibility thresholds and the rapid performance drop-off predicted.

**Compliance With Llm Reviewing Policy:**

Affirmed.

**Final Justification:**

My concerns have been addressed with clarifying notes; however some seem too trivial to be acceptable, e.g., in the case of hard prompts. Therefore, I raise my score accordingly.

**Key Questions For Authors:**

1. The Mean-Field framework (Sec 2.2) assumes permutation equivariance and duplication invariance, yet positional encodings (Definition 2.2 ) explicitly break these properties. How do the theoretical via Mean-Field transformer results remain valid for architectures that utilize positional information?
2. The paper primarily focuses on greedy decoding (argmax) to define accessibility. How does this abstraction account for practical, discrete decoding strategies such as beam search or Top-K sampling, and could these potentially alter the theoretical accessibility limits established in your analysis?

**Limitations:**

Typical threats to the validity of this work needs to be discussed.

**Strengths And Weaknesses:**

Strengths:
1. Provides a rigorous theoretical foundation for previously observed empirical failures of transformers in simple sequence tasks like exact copying.
2. Effectively employs the mathematical framework of packing numbers to quantify how finite numerical precision and embedding space geometry impose hard limits on model expressivity.
3. Demonstrates high consistency between theoretical predictions and empirical results across multiple architectures—including Pythia, Llama, and Qwen—confirming that these limitations are structural and independent of model size.
4. Tightens theoretical upper bounds by incorporating more realistic support geometries, such as ellipsoids and cones, alongside empirical estimates of non-uniform decoder cell volume distributions.

Weaknesses:
1. The theoretical results rely heavily on soft-prompt inputs (Definition 2.6) to establish an upper bound, which may significantly overestimate the actual expressivity limits of discrete, hard-prompt transformers used in real-world scenarios.
2. The model assumes a uniform grid for numerical precision (Assumption 4.3), which simplifies the non-uniform nature of standard floating-point arithmetic (e.g., fp16), where the spacing between representable values depends on the magnitude of the exponent.

---

> ### Author Rebuttal · Authors · 2026-03-27
>
> We thank the reviewer for the constructive feedback.
>
>
> **The theoretical results rely heavily on soft-prompt inputs (Definition 2.6) to establish an upper bound, which may significantly overestimate the actual expressivity limits of discrete, hard-prompt transformers used in real-world scenarios.**
>
> That is correct! Note, however, that bounds for hard prompts are much easier to derive, and our results extend directly to this setting.
>
> Formally, in the proof of Theorem 4.5 we can replace $B^{d\times m}(0,r)$ with $\mathcal V^m$, and in Theorem 4.8 replace $\mathcal G$ with the set of distributions over $\mathcal V$ (or over $\mathcal V \times [0,1]$ to incorporate positional information). The resulting bounds are straightforward to compute, and we have included them in the revised manuscript.
>
> Finally, the cramming task inherently involves soft-prompt inputs, and that is why we find our bounds to be tight for this setting.
>
> **The model assumes a uniform grid for numerical precision (Assumption 4.3), which simplifies the non-uniform nature of standard floating-point arithmetic (e.g., fp16), where the spacing between representable values depends on the magnitude of the exponent.**
>
> We thank the reviewer for this valuable observation. We now provide a refined bound on the packing number that accounts for non-uniform precision in the one dimensional case. We can then extend the bound to several dimensions by writing the embedding space as a product of one-dimensional spaces.
>
> Let $\mathcal E = [r_{\min}, r_{\max}]$ be the embedding space, where $0 < r_{\min} \le r_{\max}$ (by symmetry, we can easily extend to the setting where $\mathcal E=[-r_{\mathrm{max}}^{-},-r_{\mathrm{min}}^{-}]\cup[r_{\mathrm{min}}^{+},r_{\mathrm{max}}^{+}]$).
>
> Define
> $$
> a = \lceil \log_2 r_{\min} \rceil,
> \qquad
> b = \lceil \log_2 r_{\max} \rceil.
> $$
> Then
> $$
> 2^{a-1} < r_{\min} \le 2^a,
> \qquad
> 2^{b-1} < r_{\max} \le 2^b.
> $$
>
> And
> $$
> [r_{\min}, r_{\max}]=
> [r_{\min},2^a]
> \cup
> [2^a,2^{a+1}]
> \cup \cdots \cup
> [2^{b-1},r_{\max}].
> $$
> Hence, by subadditivity of the packing number,
> $$
> \mathcal P(\mathcal E)
> \le
> \mathcal P([r_{\min},2^a])
> +\sum_{j=a}^{b-2}\mathcal P([2^j,2^{j+1}])
> +\mathcal P([2^{b-1},r_{\max}]).
> $$
>
> Using the precision at scale $2^j$,
> $
> \varepsilon_j = 2^{j-12},
> $
> we obtain
>  - $
> \mathcal P([r_{\min},2^a])
> \le
> \frac{2^{a+1}-2r_{\min}}{2^{a-12}},
> $
>  - $
> \mathcal P([2^j,2^{j+1}])
> \le
> \frac{2^{j+1}-2^j}{2^{j-12}}
> =2^{12},
> $
>  - $
> \mathcal P([2^{b-1},r_{\max}])
> \le
> \frac{2r_{\max}-2^b}{2^{b-12}}.
> $
>
> Therefore,
> $$
> \mathcal P(\mathcal E)
> \le
> \frac{2^{a+1}-2r_{\min}}{2^{a-12}}
> +
> 2^{12}(b-a-1)
> +
> \frac{2r_{\max}-2^b}{2^{b-12}}.
> $$
> After simplification,
> $$
> \mathcal P(\mathcal E)
> \le
> 2^{12}(b-a)+2^{13-b}r_{\max}-2^{13-a}r_{\min}.
> $$
>
> Thus, we obtain the upper bound
> $$
> \boxed{
> \mathcal P([r_{\min},r_{\max}])
> \le
> 2^{12}(b-a)+2^{13-b}r_{\max}-2^{13-a}r_{\min}
> }.
> $$
>
> These updated bounds induce changes of $\pm$50\% in the numerical results.
>
> **The Mean-Field framework (Sec 2.2) assumes permutation equivariance and duplication invariance, yet positional encodings (Definition 2.2) explicitly break these properties. How do the theoretical via Mean-Field transformer results remain valid for architectures that utilize positional information?**
>
> First, as noted in Remark A.4, the mean-field framework can be extended to masked attention by augmenting the support of the probability measures to $[0,1] \times \mathbb{R}^d$. The additional $[0,1]$ coordinate encodes token positions, following [1].
>
> As for positional encodings, our definition of a transformer (Definition 2.6) explicitly excludes this component. This is because the input space in our analysis corresponds to embeddings *after* positional encoding. Therefore, modifying the positional encoding only affects the theoretical results indirectly, for instance through changes in the embedding radius $r$. We have clarified this point in the revised manuscript.
>
> **The paper primarily focuses on greedy decoding (argmax) to define accessibility. How does this abstraction account for practical, discrete decoding strategies such as beam search or Top-K sampling, and could these potentially alter the theoretical accessibility limits established in your analysis?**
>
> A direct corollary of our results is that if a sequence is not accessible under argmax decoding, then it cannot be generated with probability greater than 50\% under any stochastic decoding strategy (e.g., beam search or Top-$K$ sampling).
>
> As a result, our findings for the copying task extend to discrete decoding strategies, as they provide an upper bound on the success probability. For the cramming task, argmax decoding is used by definition, so our analysis applies directly.
>
>
>
> [1] Castin, V., et al. (2024). How Smooth Is Attention?. ICML

---

> > ### Author Rebuttal · Reviewer_SYWu · 2026-04-03
> >
> > W1: How would the revised extension proceed, which however is not quite straightforward or effortless?

---

> > > ### Author Response · Authors · 2026-04-04
> > >
> > > We provide below the corresponding versions of our main theorems when restricting to hard prompts.
> > >
> > > **Theorem 4.5**:
> > > When $n>m$, there exist some sequences of length at most $n$ that are not accessible by a transformer with prompt length $m$. Moreover, the proportion of accessible token sequences decays exponentially fast with $n$ when it exceeds the previous threshold, with rate $|\mathcal V|^{m-n}\in O(\frac1{|\mathcal V|^n})$.
> > >
> > >
> > > **Proof**:
> > > Let us fix a transformer $\tau$ with embedding radius $r$ and precision $\epsilon$. Clearly, there are only $|\mathcal V|^m$ different inputs of size $m$. Hence, the number of distinct accessible sequences of tokens with prompt length $m$ is upper bounded by $|\mathcal V|^m$. On the other hand, the number of distinct possible output token sequences of size $n$ is given by $|\mathcal V|^n$.
> > >
> > > For **Theorem 4.8**, the natural extension to hard prompts is given by Theorem D.3. We refer the reviewer to Appendix D (line 788), where both the hard-prompt and soft-prompt versions are stated explicitly.

---

### Decision · Program_Chairs · 2026-04-30

**Decision:**

Accept (spotlight)

**Comment:**

This paper investigates expressivity properties of Transformers through a new framework involving accessible sequences, which are the set of available generations (over all prompts) for a given architecture. Precise theoretical results are provided with succinct takeaways, such as how the length of accessible output generations vary as a function of the input prompt length. The work is genuinely novel and produces interesting theoretical and empirical results. All reviewers recommend acceptance, particularly with critical changes discussed during the discussion phase such as renaming the paper's title and improving the framing of results. After a revision adding results and changes from the discussion phase, this is a strong paper that provides a meaningful addition to the literature.